# No Location Left Behind: Measuring and Improving the Fairness of Implicit Representations for Earth Data

**Daniel Cai, Randall Balestriero**
Department of Computer Science
Brown University

## Abstract

Implicit neural representations (INRs) exhibit growing promise in addressing Earth representation challenges, ranging from emissions monitoring to climate modeling. However, existing methods disproportionately prioritize global average performance, whereas practitioners require fine-grained insights to understand biases and variations in these models. To bridge this gap, we introduce FAIR-EARTH: a first-of-its-kind dataset explicitly crafted to examine and challenge inequities in Earth representations. FAIR-EARTH comprises various high-resolution Earth signals and uniquely aggregates extensive metadata along stratifications like landmass size and population density to assess the fairness of models. Evaluating state-of-the-art INRs across the various modalities of FAIR-EARTH, we uncover striking performance disparities. Certain subgroups, especially those associated with high-frequency signals (e.g., islands, coastlines), are consistently poorly modeled by existing methods. In response, we propose spherical wavelet encodings, building on previous spatial encoding research. Leveraging the multi-resolution capabilities of wavelets, our encodings yield consistent performance over various scales and locations, offering more accurate and robust representations of the biased subgroups. These open-source contributions[1] represent a crucial step towards the equitable assessment and deployment of Earth INRs.

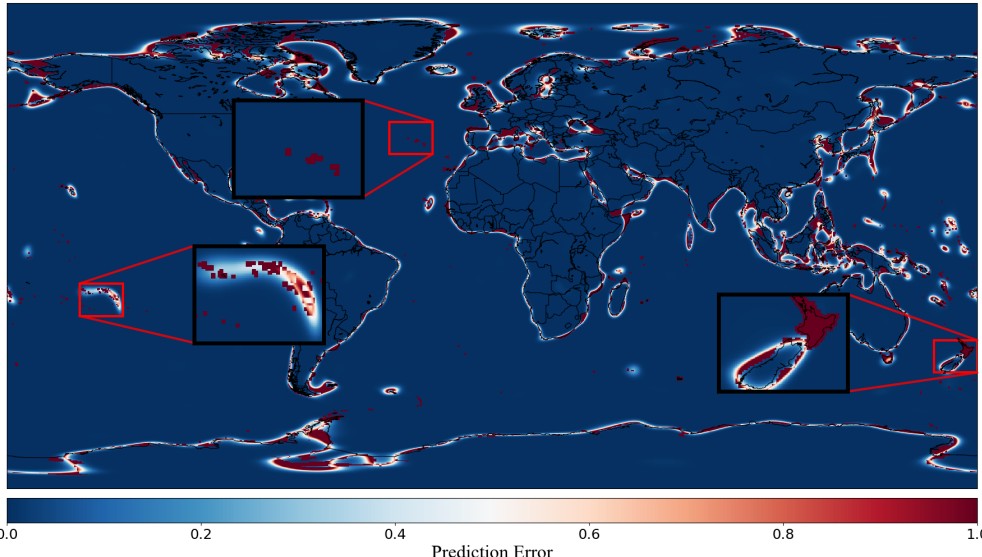

Figure 1: Heatmap of the spatial distribution of approximation errors using existing INRs to model land-sea data of the Earth. Leveraging the resolution of FAIR-EARTH, **we uncover clear bias against islands, where error magnitude is significantly higher**. Against the same task, Spherical Wavelets resolves these issues by **reconciling global signals with fine and localized signals**. Details and plots available in Appendix A.3.

---

[1]Available as part of the FAIR-EARTH package (`github.com/raccooncai448/FAIREarth`)

# 1 INTRODUCTION

Implicit neural representations (INRs) have experienced a rapid surge in popularity since the ground-breaking work of Mildenhall et al. (2020) first demonstrated their effectiveness in mapping complex 3D scenes to continuous functions. Fundamentally, early INRs were formulated as a function $f_\theta : X \to \theta$ with sinusoidal activation and weights $\theta$, where $X$ is typically a coordinate space and $Y$ is the target space (Sitzmann et al., 2020). The key enabling feature of this representation was the continuous support of $f_\theta$, yielding resolution-independent representations.

In recent years, the versatility and advantages of INRs have made them increasingly attractive across diverse fields (Molaei et al., 2023; Chen et al., 2023; Rußwurm et al., 2024). Unlike traditional discrete methods, INRs are essentially infinite-resolution models, enabling them to effectively handle signal-processing tasks involving sharp discontinuities or high variance features (Xu et al., 2022). Furthermore, they naturally excel in scenarios requiring differentiable representations, facilitating tasks such as gradient-based optimization in physics-informed machine learning (Raissi et al., 2017). The compact and flexible parameterization of INRs also offers substantial memory savings compared to grid-based methods, a critical consideration for any practitioner working with resource constraints (Liu et al., 2024).

More recently, INRs have made notable strides in Earth science applications, with transformative potential for pressing tasks such as climate modeling and environmental monitoring (Rußwurm et al., 2024; Grana et al., 2021). While existing methods span various techniques such as observation-based networks (Palecki et al., 2013), satellite-based remote sensing (Sorooshian et al., 2014), and more recently, computer-based climate simulations (Geneva and Foster, 2024), these methods suffer from some combination of discretization error, data inconsistency, and resource-intensive inference (Allen et al., 2002). In contrast, the ability of INRs to learn nonparametric models from arbitrarily high-resolution, multi-modal data presents an efficient and promising alternative. Specifically, the use of INRs to learn the underlying dynamics of explicitly *geospatial data* has rapidly garnered interest (Cole et al., 2023). These methods encode the generative process and representation of the data through an implicit function, typically a Deep Neural Network (DNN), that maps spatial coordinates to data realizations (Hillier et al., 2023).

However, as INRs transition into these realms of Earth science application, it is imperative to thoroughly evaluate the fairness and potential biases of these models. Skewed or unfair models in this domain can have far-reaching consequences, potentially amplifying existing societal inequalities or misallocating resources in vulnerable communities (Munday and Washington, 2018; Flores et al., 2022). For instance, two concrete instances of modeling bias in the Earth science and INR domains underscore the critical importance of fairness considerations:

- Federal Emergency Management Agency (FEMA) flood maps, which rely on an ensemble of machine learning models for risk assessment, have been shown to systematically underestimate flood risk in lower-income neighborhoods (Flores et al., 2022).

- Global climate models, including those used in the Coupled Model Intercomparison Project (CMIP), exhibit systematic errors in simulating precipitation patterns over Africa, consistently overestimating precipitation in southern regions while underestimating it in central regions (Munday and Washington, 2018).

While such fairness concerns should evidently be top-of-mind, as of writing, there exist *no systematic datasets* to target such biases for implicit Earth representations. Hence, we introduce the first-of-its-kind FAIR-EARTH (Fairness Assessment for Implicit Representations of Earth Data) dataset, a comprehensive framework for evaluating and mitigating biases of implicit Earth representations.

**Contribution 1:** FAIR-EARTH comprises a diverse set of benchmark datasets representing a wide spectrum of Earth science signals, from global precipitation to carbon emissions (Section 3.2). Uniquely, it also includes a suite of methodologies and metadata designed to quantify disparities in model performance across different stratifications, setting it apart from existing datasets (Table 7). In essence, FAIR-EARTH unifies disparate existing geospatial datasets under a consistent framework, enriched with extensive metadata spanning fields such as geographical features (e.g., islands, coastlines), demographic information (e.g., population density) and geopolitical boundaries (e.g., country divisions).

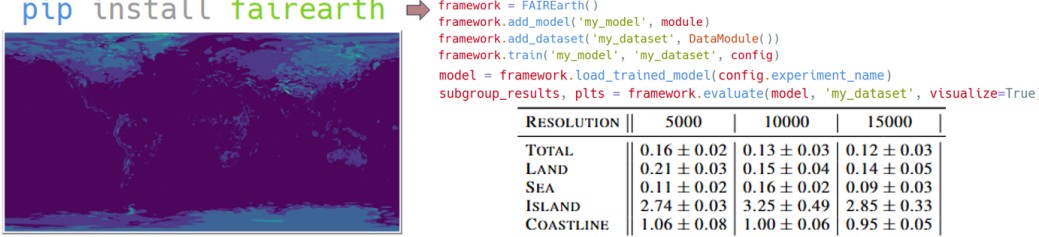

Figure 2: FAIR-EARTH is the first Earth INR-framework for fairness assessment, and moreover is an *open-source package* that allows for agile testing and analysis of subgroup-level performance.

Extensive experiments against FAIR-EARTH reveal significant disparities in the performance of state-of-the-art INR methods across different subgroups (Section 4.1). In particular, our analysis uncovers a strong negative correlation between landmass size and representation loss, with areas corresponding to localized signals and high-frequency features exhibiting consistently poor performance (Fig. 4).

To address these fairness issues, we build upon recent research into *location encodings*, which aim to simplify downstream Earth representation tasks by transforming the input into a more *learning-friendly* format. Previous work in this domain (Rußwurm et al., 2024) has focused on improving location encodings via decomposition onto the harmonic domain. While these Fourier-inspired approaches have demonstrated improvements in global accuracy and longitudinal consistency, our empirical results show that the global support of Fourier bases introduces biases against localized signals, compromising more fine-scaled geographic representations (Fig. 1).

**Contribution 2:** With these limitations in mind, we propose SPHERICAL WAVELETS (SW), a novel encoding mechanism grounded in existing research in the wavelet domain (Section 4.2). By explicitly modeling geospatial phenomena at multiple scales, SPHERICAL WAVELETS enable efficient and accurate modeling of regions with highly localized signals (Fig. 19). Our approach is motivated by the inherent multi-scale nature of Earth systems, and allows for more equitable representation across diverse geographical contexts. Against the FAIR-EARTH framework, we demonstrate that our encodings significantly reduce performance disparities compared to existing methods, paving the way for fairer Earth science applications (Fig. 6).

We summarize our contributions below, which we hope will pave the way towards more fair and equitable implicit Earth representations:

- We introduce FAIR-EARTH, a comprehensive framework for assessing fairness of implicit Earth representations across multiple modalities, enabling users to perform any fine-grained fairness study as required by their application (Section 3.2).

- We perform a thorough fairness evaluation of current INR frameworks against FAIR-EARTH. We discover significant subgroup disparities in current state-of-the-art INR methods (Section 4.1).

- We propose spherical wavelet encodings for INRs, motivated by the shortcomings of existing INRs when dealing with multi-resolution data. Against FAIR-EARTH, we demonstrate competitive performance while significantly mitigating per-group biases (Section 4.3).

## 2 BACKGROUND

**Fairness of Implicit Neural Representations** Recent work has shown that optimizing solely for average test performance can have detrimental effects on sub-group performance in natural image classification tasks (Balestriero et al., 2022; Kirichenko et al., 2023). These findings raise natural concerns about potential fairness issues in implicit neural representations (INRs), particularly in the context of Earth data. For example, algorithms fine-tuned to low-frequency global signals may overlook high-frequency fluctuations, potentially leading to biased representations. However, to the best of our knowledge, a *systematic study of fairness* in INRs for Earth data has yet to be conducted.

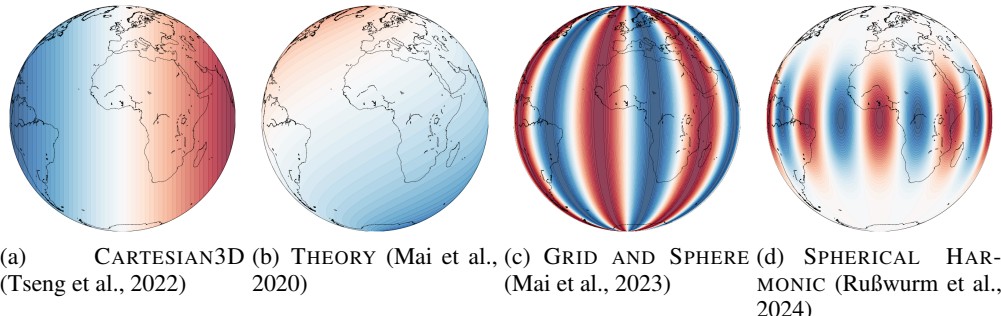

(a)  CARTESIAN3D (b) THEORY (Mai et al., (c) GRID AND SPHERE (d) SPHERICAL HAR-
(Tseng et al., 2022)  2020)  (Mai et al., 2023)  MONIC (Rußwurm et al., 2024)

Figure 3: Visualization of existing encoding mechanisms.

We focus on the work by Rußwurm et al. (2024), which set state-of-the-art benchmark performance by decomposing positional information onto spherical harmonic basis functions and integrating these encodings with SirenNets (Sitzmann et al., 2020). The core contribution is the representation of Earth data as continuous signals on the globe $f : (\lambda, \phi) \mapsto \mathbb{R}$, where $\lambda$ and $\phi$ denote longitude and latitude, respectively. For well-behaved functions with exponential decay of their eigenvalues, the signal can be precisely recovered using the following decomposition:

$$f(\lambda, \phi) = \sum_{l=0}^{\infty} \sum_{m=-l}^{l} w_l^m Y_l^m(\lambda, \phi), \tag{1}$$

where $Y$ is the class of spherical harmonic functions, $w$ are learnable scalar weights, and $l$ and $m$ are the degree and order of the basis function $Y_l^m$. Both as a practice and a necessity (Fig. 17), an upper bound on $l$ is imposed, restricting the embedding size to $l^2$ and capping the representable frequency.

**Location Encodings for INRs**  The work in Rußwurm et al. (2024) notably represents a significant advancement in the task of constructing proper *location encodings* for implicit Earth representations. In a general sense, location encodings can be viewed as a simpler alternative to embeddings, a familiar concept in natural language or multimodal learning fields. Like embeddings, encodings play a crucial role in transforming the complex task of learning the distribution $P(y|\mathbf{x})$ into a simpler task $P(y|\mathbf{x^*})$, where $\mathbf{x}$ is the location, $y$ is the predictive task, and $\mathbf{x^*}$ is the location encoding (Mai et al., 2022).

Early location encodings leveraged Gaussian processes and kernel methods (Mai et al., 2022), while more recent approaches have improved performance even with simpler mappings like DIRECT $((\phi, \lambda) \mapsto (\phi, \lambda))$ and CARTESIAN3D $((\phi, \lambda) \mapsto (x, y, z))$. As datasets have grown in dimensionality, the field has built upon variants of GRID/THEORY embeddings, which involve simple interaction terms between sine and and cosine functions on the globe. Notable examples include the GRID AND SPHERE family of encodings (SPHEREC+ and SPHEREM+) which generalizes and combines these previous encodings (Mai et al. (2023)). More recently, as INRs have gained popularity, the focus has shifted towards more complex periodic representations of geospatial data, as exemplified by the harmonic-based encodings in Rußwurm et al. (2024).

However, these global decomposition methods (Table 6) may not be accurate or efficient for common natural signals across the globe (Fig. 3). Fourier and harmonic-based methods are especially known to alias in localized, sharp, or discontinuous portions of the signal (Mallat, 1999), which intuitively can lead to biased representations of important Earth features.

**Measuring Fairness**  As described, the global approach adopted by many existing implicit Earth representations is problematic when dealing with naturally localized or discontinuous signals. Before attempting to mitigate these biases, it is crucial for practitioners to understand their nature and extent. More specifically, there is a pressing need to quantify the trade-offs in per-group performance of existing models (Garcia-Silva et al., 2018). This emphasis on group-specific performance is not merely an academic exercise but is driven by practical considerations: for tasks like natural disaster risk assessment where consequences are severe, there is a natural emphasis on improving worst-case performance rather than average-case metrics (Kemp et al., 2022).

An ideal framework for measuring fairness in Earth representations should encompass several key components: consistent high-resolution data for uniform assessment, multi-modal integration to reflect various Earth systems, and standardized fairness metrics to quantify biases along various themes. By developing such a comprehensive framework, the implicit learning and Earth science communities can work towards constructing more equitable models, aligning themselves with the growing emphasis on fairness and accountability in AI systems.

## 3    FAIR-EARTH: ALL-IN-ONE EVALUATION LIBRARY FOR EARTH DATA

The introduction of FAIR-EARTH, the first-of-its-kind dataset to target per-group performance, is highly motivated by both the natural tendencies of geospatial data and the well-studied biases of existing INRs. We first examine the specific inadequacies of existing datasets, and then outline the main contributions of the FAIR-EARTH dataset.

### 3.1    THE NEED FOR A FAIRNESS DATASET

The current landscape of Earth system datasets is not well-suited for the nuanced analyses described earlier. Commonly used datasets and benchmarks suffer from inherent limitations that can introduce or exacerbate biases. We point to two concrete examples below.

**The Shuttle Radar Topography Mission** (SRTM) land elevation model, despite its widespread use, suffers significant data gaps and inconsistencies in areas with steep terrain or dense vegetation (NASA Shuttle Radar Topography Mission, 2013). These gaps can lead to skewed representations of topographical features, potentially affecting critical applications such as flood risk assessment or infrastructure planning in vulnerable areas.

**The Visible Infrared Imaging Radiometer Suite** (VIIRS) nighttime lights dataset, often employed as a proxy for economic activity and urbanization, has limitations in detecting low levels of light emissions (Murphy et al., 2006). This shortcoming may underestimate activity in rural or underdeveloped areas, reinforcing existing biases in policy decisions.

These examples underscore a critical gap in the field: while existing datasets and Earth system models are clearly at risk of exhibiting biases across different stratifications, there is no established framework specifically designed to quantify these biases.

### 3.2    PROPOSED EVALUATION DATASET

To mitigate the limitations of some data sources highlighted in Section 3.1, FAIR-EARTH is based around several clean, accessible, and interpretable data modalities. Additionally, we collect a comprehensive suite of metadata to assess biases that may arise from both the data and representation model; this metadata lays the crucial groundwork for our novel fairness assessment of implicit Earth representations.

**Land-Sea Boundaries.** Based on NASA's Integrated Multi-satellitE Retrievals for GPM (IMERG) dataset (Huffman et al., 2014), this component contains (Fig. 7) coarse signals like continental landmasses while also providing high-resolution boundaries for fine-grained signals such as islands and coastlines.

**CO2 Emissions.**  Using data primarily from NASA's Orbiting Carbon Observatory-2 (OCO-2) satellite, this component provides ultra fine-grained and precise information into carbon emissions (OCO-2 Science Team/Michael Gunson, Annmarie Eldering, 2020; O'Dell et al., 2018; Taylor et al., 2023). OCO-2 is the second high-precision $CO_2$ satellite, and inaccuracies due to reflectance or cloud cover are mitigated via data assimilation. This high-resolution emissions data (Fig. 13) is crucial for environmental justice applications, for instance enabling researchers to identify localized pollution hotspots, or investigate correlations between emissions patterns and health disparities.

**Precipitation and Temperature.**  Derived from the CHELSA (Karger et al., 2017) dataset, this component provides an assortment of coarse and ultra fine-grained signals, as well as high resolution along the time dimension (Fig. 10). This temporal granularity allows for analysis of both long-term climate trends and short-term weather patterns.

**Population Density.** Leveraging the Gridded Population of the World, Version 4 (GPWv4) population dataset (Center for International Earth Science Information Network - CIESIN - Columbia University, 2018b), which integrates censuses, population registers, and spatial distributions, we synthesize population density data for each point in the grid (Fig. 12).

We emphasize FAIR-EARTH is a *unified* framework. All data is sampled and projected onto a uniform 0.1° x 0.1° grid, a state-of-the-art resolution for fine-grained analysis. Moreover, we propose attributes and metadata including landmass size, coast distance, and population density for each location, enabling us to disentangle the global prediction performance into meaningful subgroups (data specifications available in Appendix A.1 and Appendix A.1). In particular, our metadata comprises two main subgroups. First, *geographical features* are composed: based on a combination of existing data and ad-hoc definitions (Appendix A.1), we label relevant subgroups including islands, coastlines (stratified by distance), land, and sea. Additionally, we incorporate metadata on *population density* and *administrative boundaries*.

While core geospatial data is derived from existing datasets, we highlight the differences between FAIR-EARTH and existing stand-alone datasets (Table 7). Unlike datasets such as NaturalEarth (Nat) that emphasize geographic features, we combine multiple dimensions of data (environmental, demographic, and emissions) within one integrated format, allowing practitioners to model and assess the interdependence of multiple modalities. In contrast to existing climate datasets such as ERA5 (Hersbach et al., 1999), our environmental signals incorporate interconnected emissions data and metadata, crucial for nuanced investigation of environmental justice issues.

### 3.3 FAIR-EARTH LIMITATIONS

While the design choices for FAIR-EARTH facilitate subgroup-level fairness evaluations, they also yield certain tradeoffs. In particular, temporal and polar biases may manifest from the gridded format of FAIR-EARTH data; we later discover (4.3) that this format indeed induces bias in the training data. Given the agile framework of FAIR-EARTH, we strongly encourage practitioners to evaluate and incorporate their own datasets when necessary, and to adopt best practices in sampling to mitigate such biases.

## 4 MEASURING AND IMPROVING THE (UN)FAIRNESS OF IMPLICIT NEURAL REPRESENTATIONS

### 4.1 EVALUATION OF EXISTING REPRESENTATIONS

**Global Performance is not Representative of Local Performance**   To properly understand the behavior of existing INRs, we depart from hyperparameter fine-tuning, and instead perform extensive training over a cross-product of each model's hyperparameter space. This allows us to move past "best-case" analysis, and instead incorporate nuanced analysis of model tradeoffs. Moreover, as downstream tasks often require further tuning regardless, it is natural to instead consider the overall behavior of the implicit representation. Experimental details are available in Appendix A.2.

In particular, we explore the correlations of subgroup performance. While certain disparities in performance are intrinsic to the task (e.g., islands and coastlines exhibit higher representation loss as land-sea boundaries are harder to learn than constant signals), a *fair* model should in general show *concurrent improvement in subgroups*. Leveraging the metadata in FAIR-EARTH, we compare representation performance between areas that correspond to local signals (e.g., islands) and ones that correspond to global signals (e.g., large landmasses).

Strikingly, our correlation analysis indicates a stark disparity in representation loss between global and local groups among *all state-of-the-art encodings* (Appendix A.4, Section 4.1). For SPHERICAL HARMONICS, we notice a strong negative correlation between land and island loss when stratified along training resolution (Section 4.1), suggesting that when optimized for total loss, state-of-the-art INRs exhibit a clear tradeoff between global and local performance. Regardless of stratification, INRs of the form Eq. (1) appear incapable of *competitive* performance across all subgroups at once (Table 2).

Table 1: Land-island performance correlations across encodings and training resolutions. While existing encodings compromise local-global performance, SPHERICAL WAVELET reconciles improvement over multi-scale features. We omit SPHEREC+ due to abnormally high bias.

| Training Samples | Pearson Correlation (R) | | |
|---|---|---|---|
| | SW (Ours) | SH | THEORY |
| 5000 | 0.51 | −0.73 | −0.53 |
| 10000 | 0.42 | −0.88 | −0.62 |
| 15000 | 0.04 | −0.88 | −0.10 |
| 20000 | 0.31 | −0.68 | −0.28 |

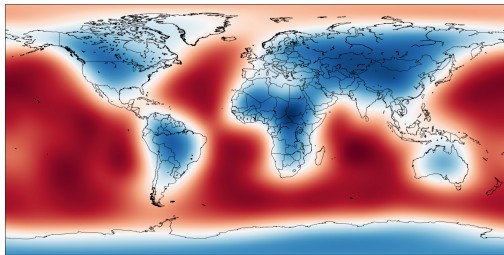 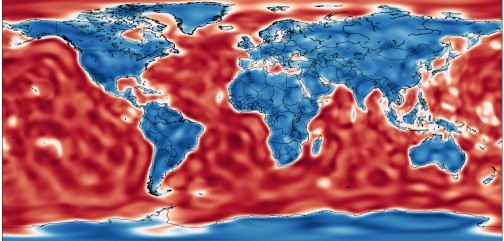

a) High regularization *implies* low resolution    b) Low regularization *implies* aliasing

Figure 4: Model behavior at different resolutions and regularizations. Smaller models fail to capture fine/local signals. Larger models poorly reconcile local signals with existing global signals.

**Aliasing as a Result of Overfitting**    As described earlier, Eq. (1) is global in nature. We observe the implications of this construction in Fig. 4, where the smoothness of land and sea signals are compromised in an effort to represent islands and coastlines. We corroborate this observation with an ablation over encoding size, which shows increasing encoding size results in plateauing land performance and deteriorating island performance (Fig. 21).

Concretely, the observed aliasing in the high-resolution case is indicative of the Gibbs phenomenon, exhibiting unnatural oscillations around the island and coastline 'discontinuities' (Dyke and Dyke, 2001). In the effort to precisely fit the islands, Eq. (1) is forced to learn an *unnatural* representation of the land and sea masses.

Our correlation analysis and aliasing investigation confirms that no SPHERICAL HARMONIC-based model is able to maintain competitive performance in both sub-groups simultaneously, suggesting that current solutions in implicit Earth representations require further development to reach truly equitable predictions.

**Biases Across Multiple Modalities and Subgroups**    While the land-sea binary classification task reveals natural biases against fine and localized areas, we emphasize that one of FAIR-EARTH's main strengths is in its ability to easily quantify subgroup disparities across multiple modalities. In particular, we perform similar stratified evaluation against FAIR-EARTH's benchmarks for environmental signals. For the surface temperature dataset, which exhibits similar sharp variations across recognizable boundaries, we note similar trends (Fig. 5). Namely, FAIR-EARTH reveals systematic patterns in representation quality: regions with sharper variations, particularly near the coast, show significantly higher average representation loss ($MSE_{Land} = 0.87$, $MSE_{Coast} = 0.101$) compared to regions with smoother variations ($MSE_{Sea} = 0.43$, $MSE_{Island} = 0.49$) (Table 14).

Moreover, this analysis extends naturally to downstream biases through FAIR-EARTH's rich metadata. At the country level, we observe that representation challenges at the feature level manifest as systematic performance disparities. For instance, SPHERICAL HARMONIC and THEORY encodings particularly struggle with to demarcate the Spain's fine Mediterranean coastline, while all studied encodings show degraded performance in coastal countries due to sharp temperature gradients at land-sea boundaries (15).

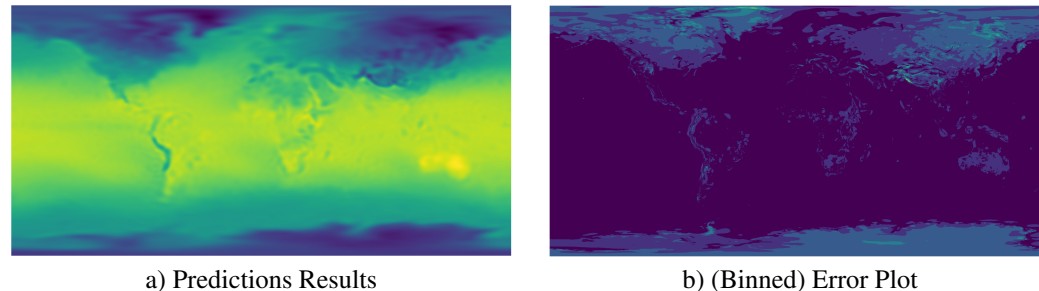

a) Predictions Results        b) (Binned) Error Plot

Figure 5: Qualitative performance of SPHERICAL HARMONIC on surface temperature regression task. SH exhibit bias against high-frequency details, with loss concentrated in areas of abrupt change (e.g., coastlines).

## 4.2 SPHERICAL WAVELETS FOR FAIR EARTH REPRESENTATIONS

As explored in Section 2, a major limitation of SPHERICAL HARMONIC was their global support, which struggled to reconcile localized signals with global ones. In particular, SH required a large amount of basis functions to properly represent localized signals, and this excessive parameterization resulted in aliasing in other regions. The ad-hoc solution involving re-weighting subgroups depends on extensive labels; moreover, the effect on global performance is unknown (Stone et al., 2024).

To this end, we are motivated to leverage the theoretical guarantees of wavelets, which allow for multi-scale resolution analysis of signals, efficiently representing signals at various scales and locations. To extend towards Earth data, we refer to past work in Demanet and Vandergheynst (2003) that introduces spherical wavelet basis functions. Similar to the formulation in Eq. (1), any function on the sphere can also be approximated by a discrete sum of wavelet basis functions, where coefficients are similarly learned via the INR mechanism.

The following sections briefly outline the construction of a novel encoding mechanism SPHERICAL WAVELET, and its seamless integration into the currently studied INR pipeline. We direct interested readers towards McEwen et al. (2007); Demanet and Vandergheynst (2003) for analysis on the correctness of the construction, and towards A.5 for evidence that SW encodings are surprisingly more *efficient* and *stable* than their SH counterparts.

**Spherical Morlet Wavelet via Inverse Projection** A concise, computationally tractable construction for spherical wavelets arises from inverse stereographic projection of Euclidean wavelets (Demanet and Vandergheynst, 2003). We note that due to the nature of this projection, the SPHERICAL WAVELET is ill-defined at the poles.

As proven in Sanz et al. (2006), this inverse stereographic projection $\Pi^{-1}$ of an admissible *Euclidean* wavelet yields an admissible *spherical* wavelet; that is, the projected wavelet $\psi(\theta, \varphi)$ satisfies the necessary zero-mean condition (Demanet and Vandergheynst, 2003). Empirically, we find that the Morlet mother wavelet shows consistently superior performance (Fig. 25). Thus, applying the inverse projection onto the 2D Morlet wavelet with width factor and wave number $w, k = 1$ yields the admissible spherical Morlet mother wavelet:

$$\psi_M(\theta, \phi) = [\Pi^{-1}\psi_{\mathbb{R}^2}](\theta, \phi) = \frac{e^{i \tan(\theta/2)\cos(\phi)}e^{-(1/2)\tan^2(\theta/2)}}{1 + \cos\theta} \tag{2}$$

**Lifting Scheme** From an admissible mother wavelet, we can now construct a wavelet basis from affine transformation on the sphere. Analogous to time-frequency localization of Euclidean wavelets, spherical wavelets provide rotation-dilation localization. The rotation operator $\mathcal{R}(\rho) \equiv \mathcal{R}(\alpha, \beta, \gamma)$ and dilation operator $\mathcal{D}(a)$ both transport a function from and into $f \in L^2(\mathbb{S}^2, d\mu(\theta, \phi))$; refer to McEwen et al. (2007) for rigid formulation. Applying these transformations, we then define a set of orthogonal basis functions $\{\psi_{a,\rho} \equiv \mathcal{R}(\rho)\mathcal{D}(a)\psi_M\}$

Finally, we discretize the reconstruction formula. First, we deterministically distribute $\rho$ across $N$ points on the sphere following the Fibonacci lattice $\mathcal{F}_N$ (González, 2009). To discretize rotation, we take $a \in \{2^{\frac{i}{Q}}, 0 < i \leq M\}$, where $Q \leq 8$ is a fixed value (Mallat, 1999). This yields an embedding

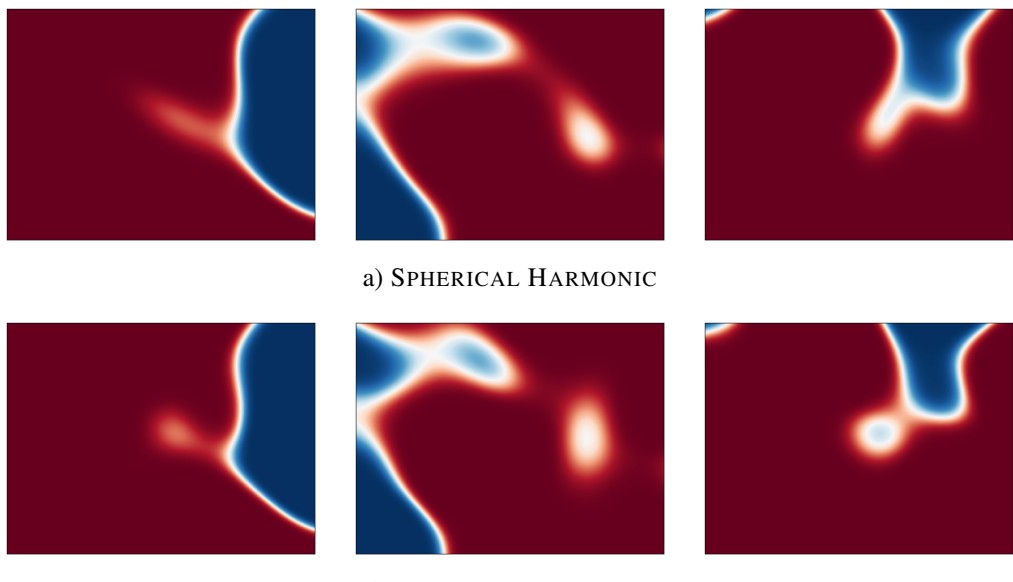

a) SPHERICAL HARMONIC

b) SPHERICAL WAVELET

Figure 6: Comparisons of SPHERICAL HARMONIC and SPHERICAL WAVELET behavior in localized regions. Full figures available in Fig. 18 and Fig. 19.

size on the order of $\mathcal{O}(NM)$, and the discretized approximation

$$f(\theta,\phi)_{N,M} \equiv \sum_{\rho \in \mathcal{F}_N} \sum_{i=1}^{M} w_{a,\rho}\psi_{a,\rho}(\theta,\phi)$$

**Localization Properties** The spherical Morlet wavelet, by design, exhibits dual localization in both dilation and rotation, offering significant advantages in signal representation on the sphere (McEwen et al., 2007). These properties enable a more nuanced and adaptive analysis of spherical data. In particular, the parameterization of the Morlet variant is parameterized is illustrated in Fig. 23.

### 4.3 SPHERICAL WAVELET EXPERIMENTS

**SPHERICAL WAVELET Corrects Localized Biases** To examine the biases present, we begin by analyzing the trends in training and evaluation. In particular, we leverage FAIR-EARTH and deploy a large cross-product of parameters ($N = 984$) to cross-validate algorithm performance between local signals and global signals over various configurations. As a baseline, we compare to results to [SPHERICAL HARMONIC, THEORY, and GRID AND SPHERE] encodings trained under identical settings, therefore isolating the effects of the encodings.

We observe a sharp contrast between the local-global correlation of our encodings; that is, SPHERICAL WAVELET learns to *efficiently* represent islands even at coarser resolution, therefore inducing a positive trend between global (land) and localized (island) performance, whereas our baseline encodings exhibit a tradeoff between global/local performance even for finer resolutions Section 4.1.

Specifically, in contrast to SPHERICAL HARMONIC, where finer features are often blended together (Fig. 1), SPHERICAL WAVELET has a significantly higher resolvable resolution for similar parameter counts. Especially in localized areas such as islands, our encodings tend to produce sharper, more fine boundaries compared to spherical harmonic encodings (Fig. 6).

**SPHERICAL WAVELET is Competitive with Existing Methods** While our correlation analysis showed performance of SPHERICAL WAVELET exhibited fair and stable *trends*, we also seek to show performance is competitive with existing methods. First, we re-examine the previous experiment and compare general global performance of SPHERICAL WAVELET and SPHERICAL HARMONIC across well-defined hyperparameter spaces.

Table 2: Comparison of best/average cross-entropy loss between SH and SW over land-sea cross-validation. Top encodings per stratification are bolded.

| Samples | SPHERICAL WAVELET | SPHERICAL HARMONIC |
|---|---|---|
| 5000 | 0.1404 / **0.1473** | **0.1392** / 0.1524 |
| 10000 | **0.1111 / 0.1169** | 0.1137 / 0.1218 |
| 15000 | **0.1000 / 0.1056** | 0.1003 / 0.1086 |
| 20000 | 0.0912 / **0.0962** | **0.0892** / 0.0969 |

In general, we observe that the learned representation is similar to that of spherical harmonics, as indicated by the visual similarities in their learned representations (Fig. 6). However, SW in fact exhibits *competitive or superior* performance over SH through various training resolutions: we maintain significantly superior average performance across all training resolutions (Table 2).

The degradation in performance for SPHERICAL WAVELET for higher resolutions may be attributed to the misspecifications of certain parameters in our model (e.g. wave number, scaling factors). To this end, we also evaluate fine-tuned performance on the temperature and precipitation regression tasks, observing more consistent performance: when properly parameterized, SPHERICAL WAVELET performs roughly on par with SPHERICAL HARMONIC encodings across all FAIR-EARTH tasks (Appendix A.4).

**SPHERICAL WAVELET Tradeoffs** While SPHERICAL WAVELET demonstrates robust performance across various tasks, like all signal-based methods, it faces certain limitations. We investigate these limitations using two challenging external datasets:

- On the checkerboard classification task Rußwurm et al. (2024), which features coarse, single-scale signals, SPHERICAL WAVELET performs well relative to most baselines but consistently falls short of SPHERICAL HARMONIC. This suggests that the wavelet decomposition may introduce unnecessary complexity for simple, single-scale tasks.

- In the land-sea classification task Rußwurm et al. (2024), which uses non-gridded, ultra high-resolution data, SPHERICAL WAVELET only marginally outperforms SPHERICAL HARMONIC. Our latitudinal analysis (Fig. 22) reveals that this limited improvement stems primarily from degraded performance near the poles.

## 5 CONCLUSION

Measuring the fairness of implicit Earth representations is pressing yet lacking field in both the Earth science and implicit representation fields. Our contributions in measuring and improving certain biases in state-of-the-art INRs represents a step forward in this important direction. By introducing FAIR-EARTH, a novel framework for high-resolution assessment of subgroup-level performance, we aim to develop and assess models pertinent to the needs of practitioners, informing of existing biases and limitations of Earth models. Leveraging this new framework, our comprehensive assessment uncovers striking biases against certain subgroups across state-of-the-art INRs. Based on these observed performance disparities, we propose SPHERICAL WAVELET, injecting localized location encodings that are naturally motivated. Against the many benchmarks of FAIR-EARTH, we resolve many of the biases evident in SPHERICAL HARMONIC, while maintaining competitive performance. We invite iteration and updates on FAIR-EARTH to improve the coverage of subgroups and signals, and stay relevant with changes and trends within the community. Additionally, given the limitations of SPHERICAL WAVELET, we hope further research can develop heuristics or structural changes to address these shortcomings; in particular, we point to the integration of non-projection based SPHERICAL WAVELET constructions as a promising direction for future work Sanz et al. (2006).

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

## A  APPENDIX

### A.1  FAIR-EARTH DETAILS

Table 3: FAIR-Earth Components

| Category | Description | Misc. | Source |
|---|---|---|---|
| Land-Sea | Binary and continuous data, based on bathymetric data on percent water surface coverage for every grid point. | Metadata includes: islands, coastlines, landmass sizes, coast distance. | (Huffman et al., 2014) |
| Population | Population density data, derived from combination of spatial distribution with national census and population register data. | Errors in Egypt and Greenland are flagged, and tentatively smoothed via nearest-neighbor interpolation. | (Center for International Earth Science Information Network - CIESIN - Columbia University, 2018a) |
| Precipitation | Global accumulated precipitation, based on composite of measurements and modeling. | Monthly Resolution | (Beck et al., 2019) |
| Temperature | Air surface temperature, based on satellite measurements and reflectance modeling. | Monthly Resolution | (Karger et al., 2017) |
| Emissions | Measured and modeled CO2 emissions from the OCO-2 satellite. | Daily Resolution | (Dou et al., 2023) |

**Metadata Details**   In particular, for subgroups with ill-defined thresholds (e.g., classification of islands and coastlines), we provide flexibility with adjustable thresholds. This feature allows researchers to fine-tune their analyses based on specific definitions or needs, which can vary depending on the research question or application domain. For the purposes of our analyses, islands are defined as landmasses with size under $30,000$ sq. miles, encapsulating most of the "minor islands" as defined by Depraetere et al. (2007).

Second, we incorporate *demographics and geographical boundaries* leveraging the Gridded Population of the World, Version 4 (GPWv4) population dataset (Center for International Earth Science Information Network - CIESIN - Columbia University, 2018b), which integrates censuses, population registers, and spatial distributions, we synthesize population density data for each point in the grid. Then, leveraging high-resolution NaturalEarth (Nat) data, we label each grid point with appropriate country labels. These labels are essential for equal representation of both densely and sparsely populated areas, preventing the overrepresentation of urban centers or the underrepresentation of rural regions, a common bias seen in survey and modeling tasks.

### A.1.1  DATASET FIGURES

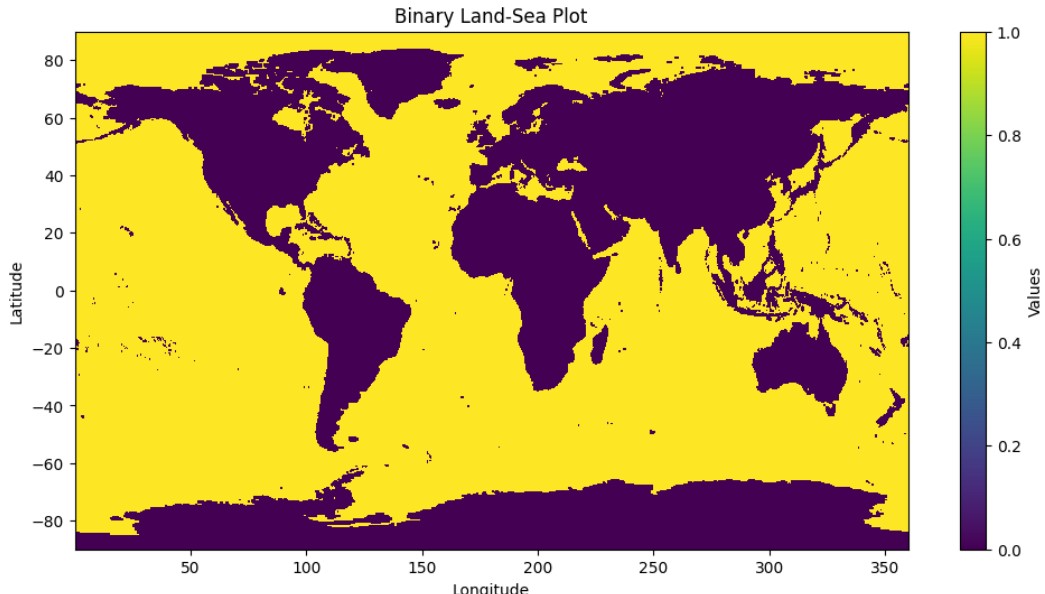

Figure 7: Binary land-sea

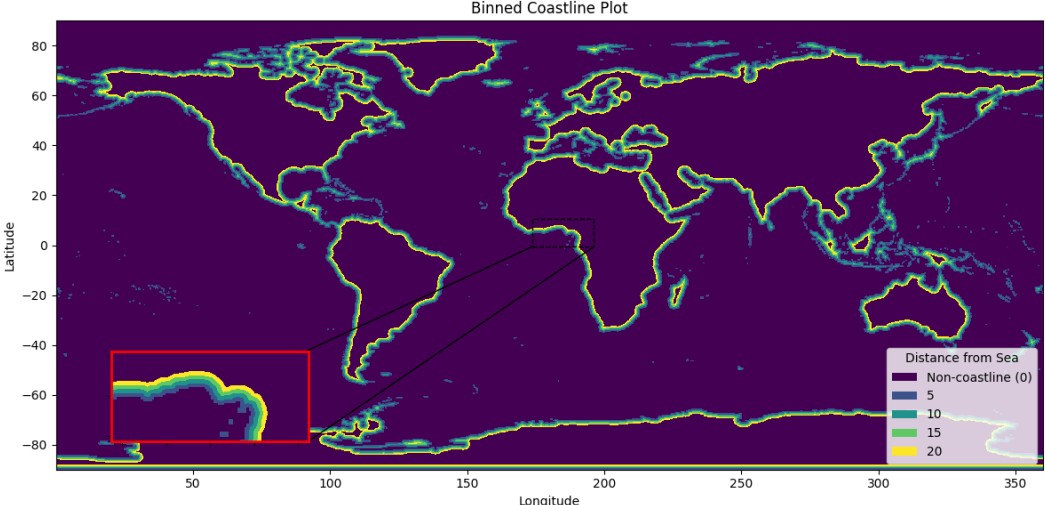

Figure 8: Coastline

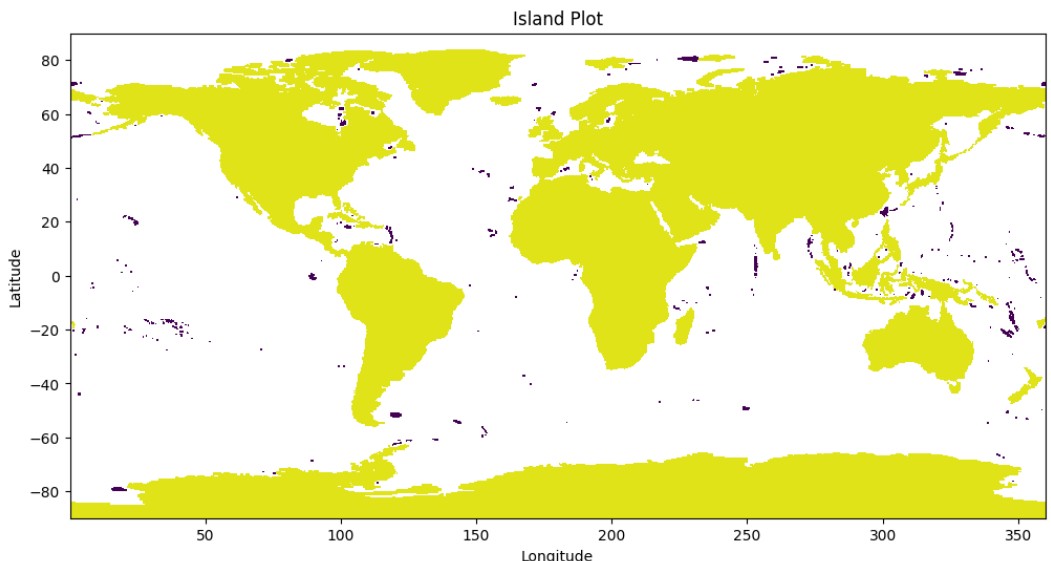

Figure 9: Islands (in dark purple)

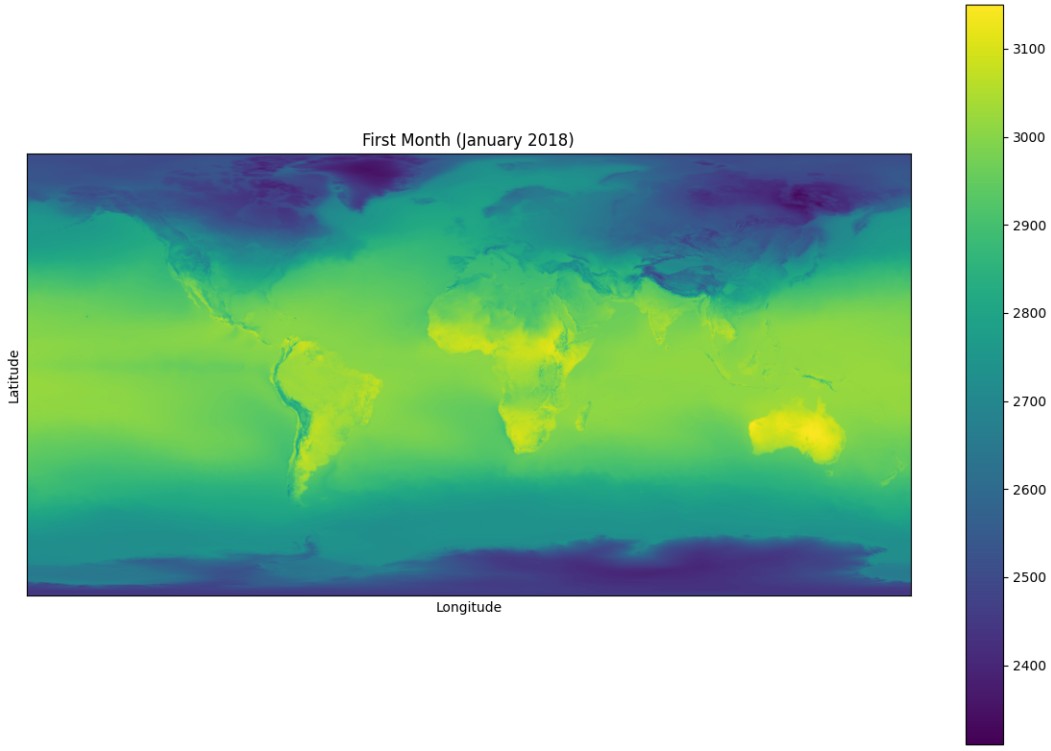

Figure 10: Air surface temperature plot (Jan. 2018)

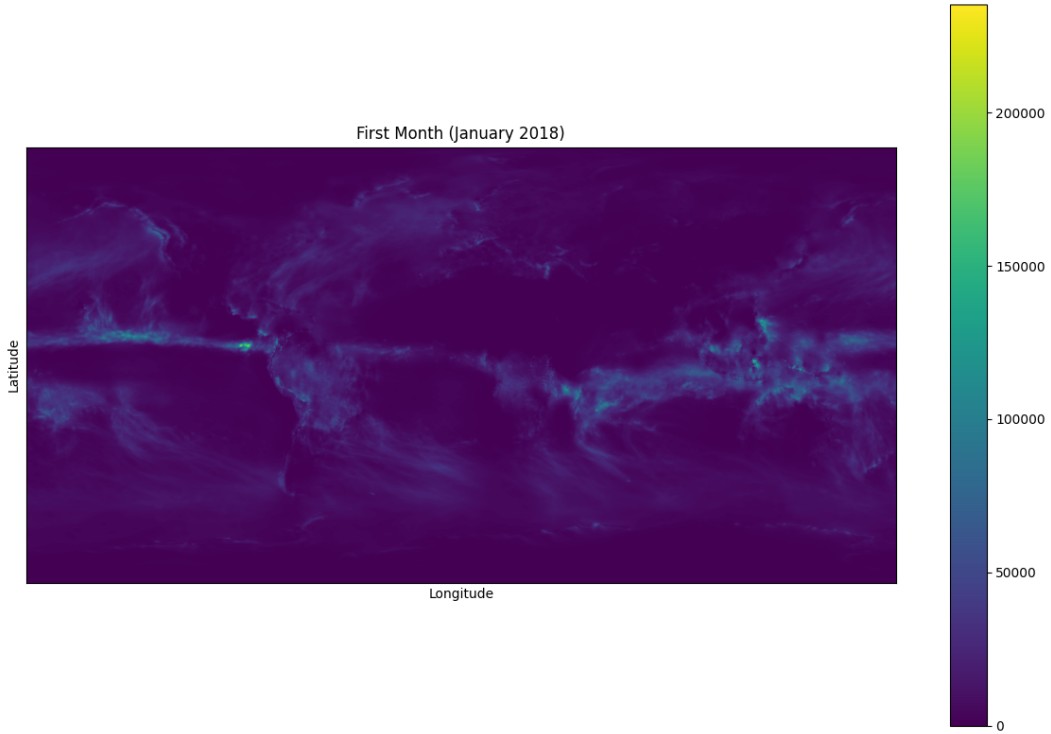

Figure 11: Cumulative precipitation plot (Jan. 2018)

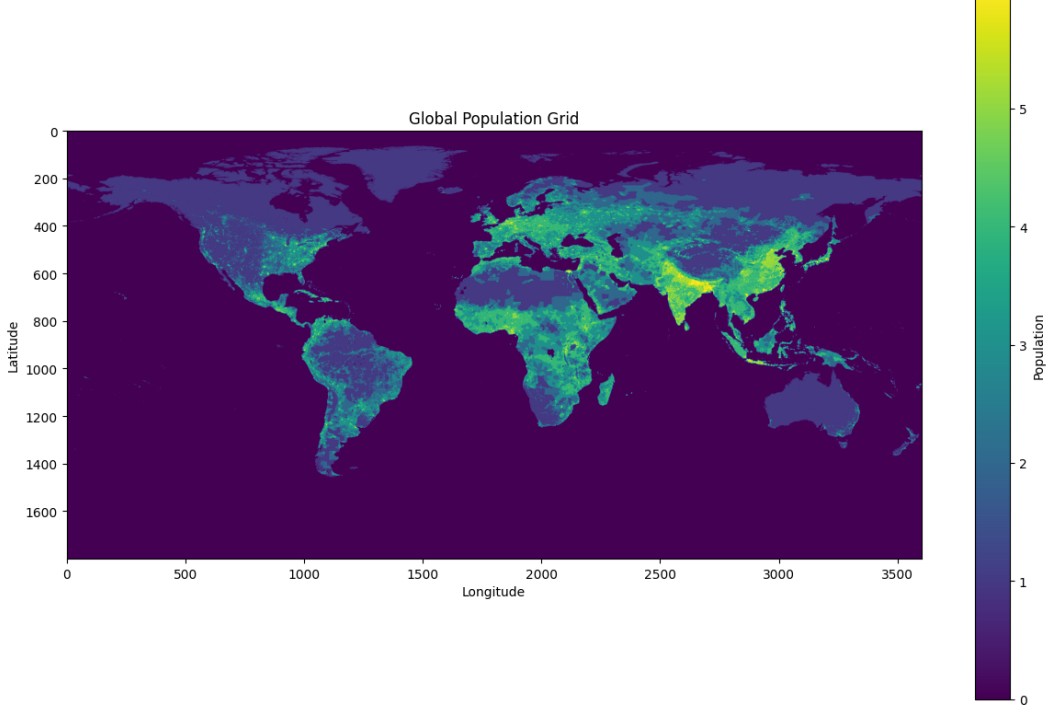

Figure 12: Global population plot

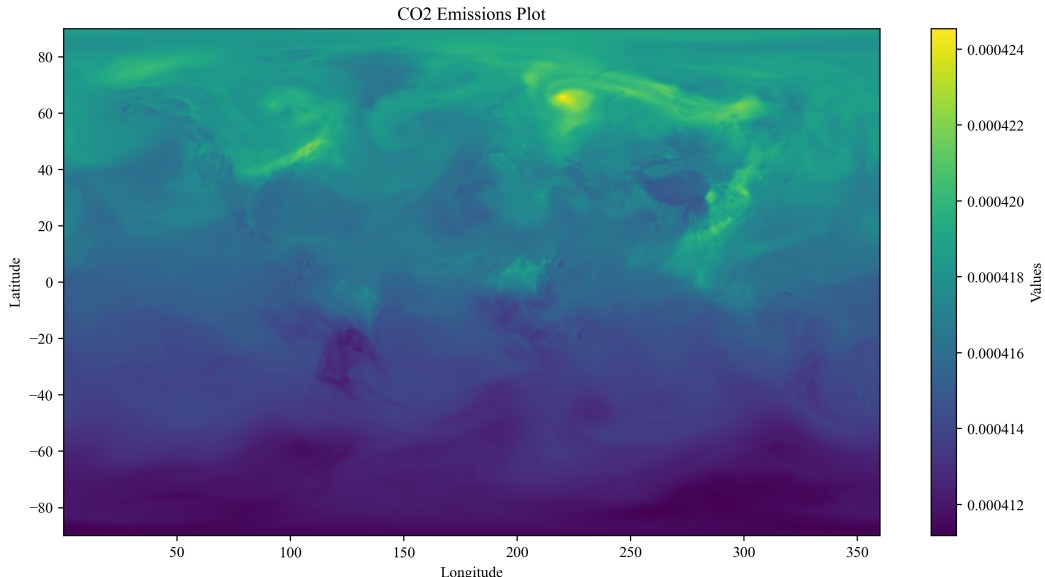

Figure 13: CO2 emissions plot

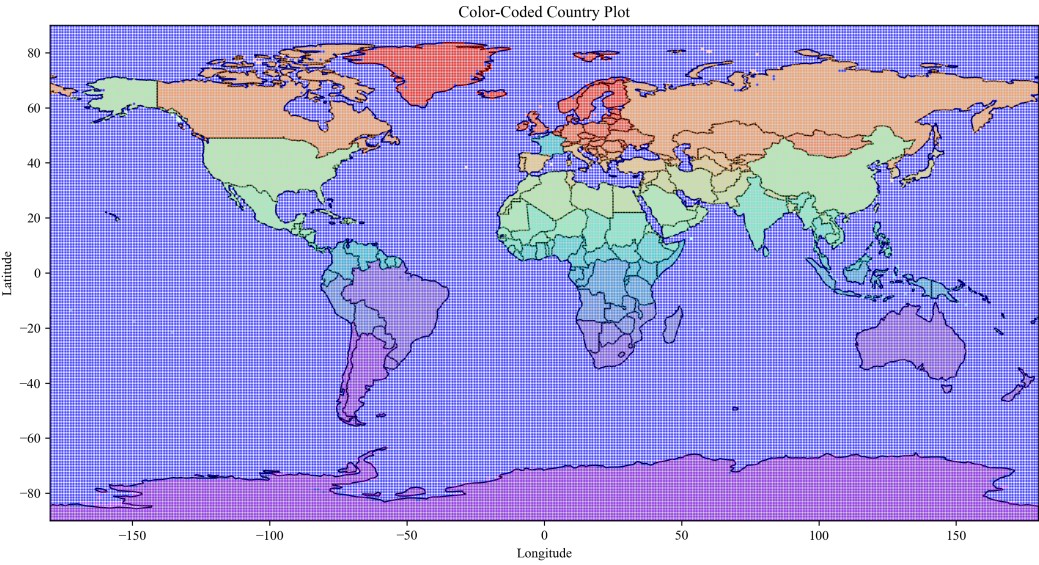

Figure 14: Color-coded country plot

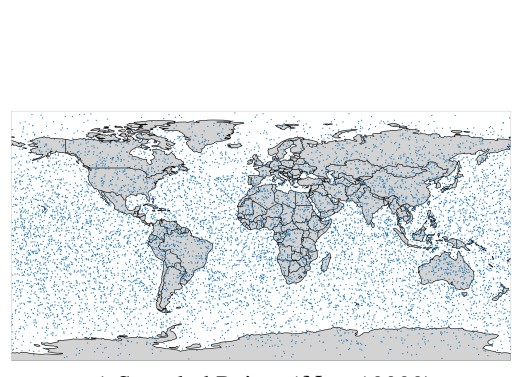

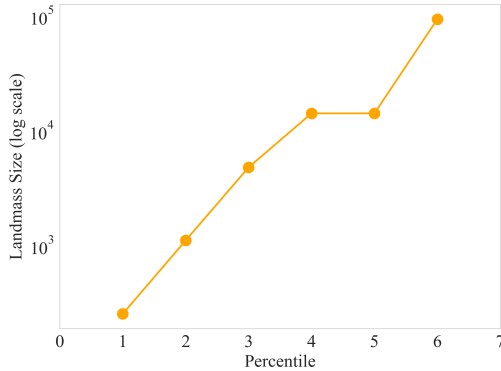

a) Sampled Points ($N = 10000$)      b) Landmass Size Percentiles for $N_{\text{land}} = 3233$

Figure 15: Coverage of different landmass sizes using uniform sampling. At $N = 10000$, at least 64 points are sampled on landmasses of size less than $40,000$ square miles, roughly the size of the main island of Cuba.

Table 4: Grid search and encoding parameters

| Parameter | Values | Applicable Encoding |
|---|---|---|
| Training Samples | {5000, 10000, 15000, 20000} | Both |
| Weight Decay | {1e-5, 1e-4, 1e-3} | All |
| Maximum Scale | {3, 4, 5} | SW |
| Maximum Rotations | {50, 90, 130, 170} | SW |
| Legendre Polynomials | {5, 7, 10, 12, 15, 17, 20, 22, 25} | SH |
| K Value | 6 | SW |
| Scale Factor | 1 | SW |
| Minimum Radius | {45, 90} | THEORY, GRID AND SPHERE |
| Frequency Numbers | {16, 32, 64} | THEORY, GRID AND SPHERE |
| Batch Size | 2048 | All |
| Learning Rate | 1e-4 | All |
| Maximum Epochs | 500 | All |

## A.2 TRAINING SPECIFICATIONS

**Land-Sea Binary Classification Training**    We conducted a comprehensive ablation study to examine model trends on the land-sea binary classification task. The study was performed using a grid-search, and relevant parameters are available in Table 4. Note that the parameter sizes for encodings were chosen to be similar, to control for any effects from simple increase in embedding space.

To initialize training, we set identical data generation seeds for each model. Validation data was sampled similar to Fig. 15 , but with $N_{\text{validation}} = 0.2 * N_{\text{training}}$ and different initialization seeds.

For fine-tuning, we perform a similar grid search over a similar continuous range of variables. The specific parameters are shown in Table 5. Due to the extra parameterization of SPHERICAL WAVELET, we quadruple the number of trials to 120, compared to 30 for all other encodings (Fig. 20).

**Precipitation and Temperature Training**    Training over these environmental signals was nearly identical to training on the land-sea binary classification task. We note a couple of minor discrepancies:

- Mean-squared error loss was applied to a normalized version of the data, as opposed to binary cross-entropy.

- For the sake of precision and brevity, all precipitation losses in the paper are increased by a magnitude of 10. We emphasize that this doesn't affect our analyses.

Table 5: Hyperparameter ranges for model optimization

| Category | Hyperparameter | Range | Step |
|---|---|---|---|
| Architecture | Hidden Dimension | $32 - 96$ | 32 |
| | Number of Layers | $1 - 3$ | 1 |
| SPHERICAL WAVELET | Maximum Scale | $2 - 5$ | 1 |
| | Maximum Rotations | $20 - 200$ | 40 |
| | K Value | $4 - 10$ | - |
| | Scale Factor | $0.75 - 1.25$ | - |
| SPHERICAL HARMONIC | Legendre Polynomial Degree | $10 - 30$ | - |
| Training | Learning Rate | $10^{-4} - 10^{-1}$ | log scale |
| | Weight Decay | $10^{-8} - 10^{-1}$ | log scale |

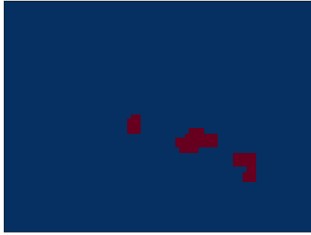 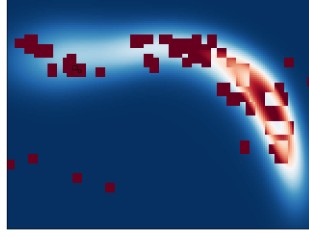 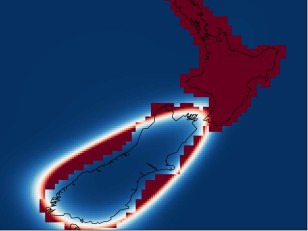

Figure 16: Zoomed-in inset plots for main diagram

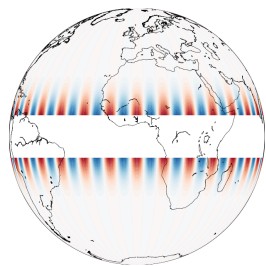

Figure 17: At higher frequencies ($L = 30$), the closed-form calculations provided in Rußwurm et al. (2024) run into numerical error.

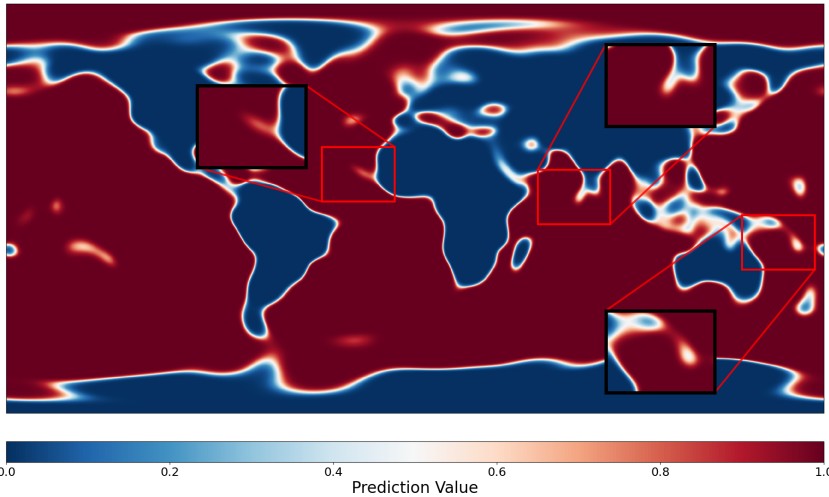

Figure 18: SPHERICAL HARMONIC encoding prediction plot

## A.3 MISCELLANEOUS FIGURES

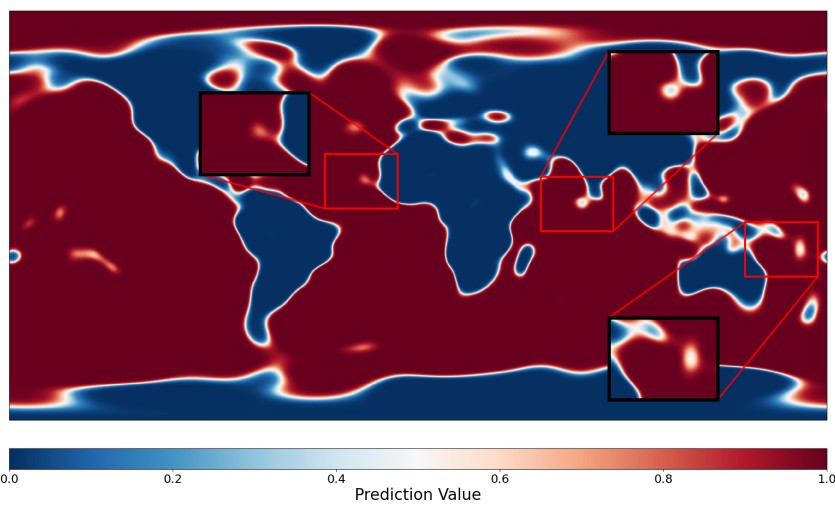

Figure 19: SPHERICAL WAVELET encoding prediction plot

Table 6: Comparison of existing location encodings

| Encoding | Resolution-Adaptive | Basis Type | Localized Basis |
|---|---|---|---|
| SPHERICAL WAVELET | ✓ | Wavelet | ✓ |
| SPHERICAL HARMONIC | ✓ | Harmonic | × |
| CARTESIAN3D | × | Coordinate | × |
| THEORY | ✓ | Sinusoidal + Coordinate | × |
| GRID AND SPHERE | ✓ | Sinusoidal + Coordinate | × |

Table 7: Comparison of common geospatial datasets

| Dataset | Multimodal | Temporal Resolution | Metadata |
|---|---|---|---|
| FAIR-EARTH | ✓ | ✓ | ✓ |
| NaturalEarth | ✓ | × | × |
| ERA5 (Hersbach et al., 1999) | × | ✓ | ✓ |
| OpenStreetMap (OpenStreetMap contributors, 2017) | × | ✓ | ✓ |

Table 8: Average subgroup cross-entropy test loss of SPHERICAL HARMONIC across cross-validation, stratified by spatial resolution. **We observe that the bias of the model in missing "island" and "coastline" persists even for high resolution dataset, as improvement plateaus.**

| RESOLUTION | 5000 | 10000 | 15000 | 20000 | 25000 | 30000 |
|---|---|---|---|---|---|---|
| TOTAL | $0.16 \pm 0.02$ | $0.13 \pm 0.03$ | $0.12 \pm 0.03$ | $0.11 \pm 0.04$ | $0.10 \pm 0.04$ | $0.10 \pm 0.04$ |
| LAND | $0.21 \pm 0.03$ | $0.15 \pm 0.04$ | $0.14 \pm 0.05$ | $0.14 \pm 0.05$ | $0.12 \pm 0.06$ | $0.16 \pm 0.06$ |
| SEA | $0.11 \pm 0.02$ | $0.16 \pm 0.02$ | $0.09 \pm 0.03$ | $0.08 \pm 0.03$ | $0.08 \pm 0.03$ | $0.80 \pm 0.03$ |
| ISLAND | $2.74 \pm 0.03$ | $3.25 \pm 0.49$ | $2.85 \pm 0.33$ | $2.66 \pm 0.26$ | $2.61 \pm 0.20$ | $2.51 \pm 0.21$ |
| COASTLINE | $1.06 \pm 0.08$ | $1.00 \pm 0.06$ | $0.95 \pm 0.05$ | $0.90 \pm 0.04$ | $0.82 \pm 0.04$ | $0.81 \pm 0.05$ |

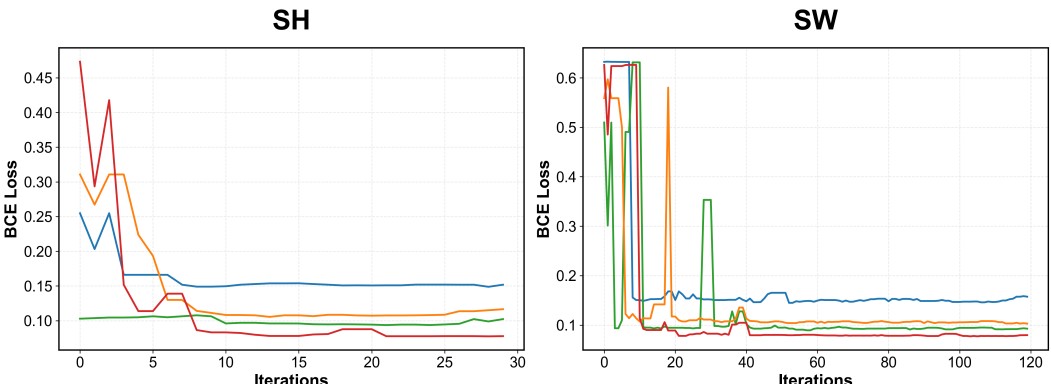

Figure 20: Fine tuning dynamics for different training resolutions, showing 5-iteration median window. Despite the larger parameter space, fine-tuning of SPHERICAL WAVELET converges at a comparable, slightly slower rate than SPHERICAL HARMONIC.

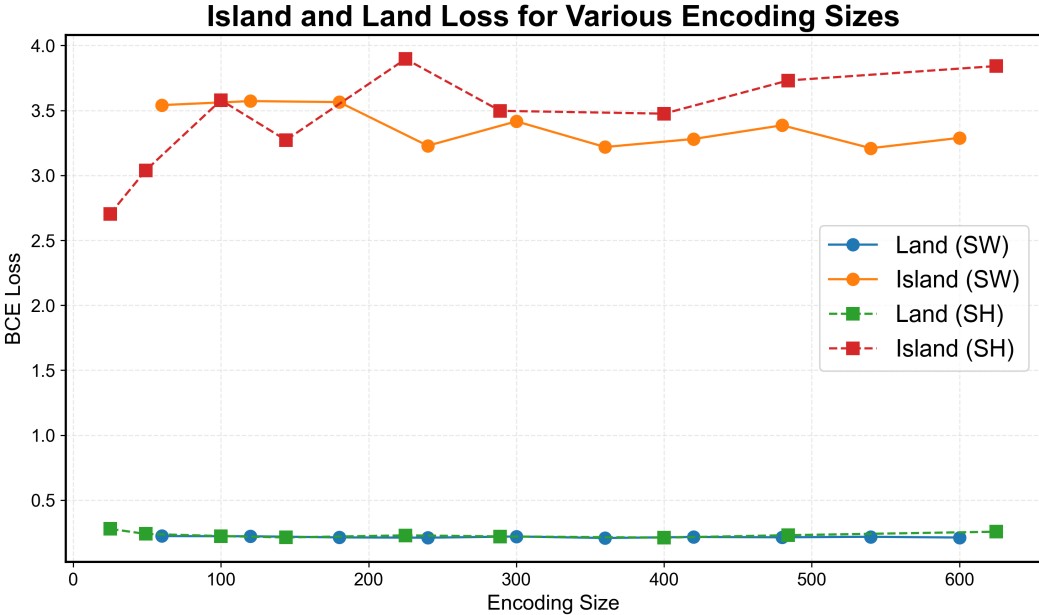

Figure 21: Land and island loss for SH, SW representations, trained on the same resolution with varying encoding sizes. The improved performance of SPHERICAL WAVELET may be attributed to better representation of these fine-scale features. Interestingly, SPHERICAL HARMONIC's biases against islands *worsen* with increased encoding size.

A.4  EXTENDED EXPERIMENTAL RESULTS

| PE ↓ Resolution → | 5000 | 10000 | 15000 | 20000 |
|---|---|---|---|---|
| SPHERICAL WAVELET | $0.1419 \pm 0.0009$ | $0.0890 \pm 0.0003$ | $0.1017 \pm 0.0007$ | $0.0762 \pm 0.0005$ |
| SPHERICAL HARMONIC | $0.1462 \pm 0.0013$ | $0.0933 \pm 0.0010$ | $0.1057 \pm 0.0010$ | $0.0767 \pm 0.0005$ |
| THEORY | $0.1586 \pm 0.0034$ | $0.1137 \pm 0.0012$ | $0.1324 \pm 0.0061$ | $0.1016 \pm 0.0073$ |
| SPHEREM+ | $0.6314 \pm 0.0006$ | $0.6307 \pm 0.0003$ | $0.6358 \pm 0.0001$ | $0.6259 \pm 0.0001$ |
| SPHEREC+ | $0.6307 \pm 0.0017$ | $0.6308 \pm 0.0002$ | $0.6357 \pm 0.0004$ | $0.6259 \pm 0.0002$ |

Table 9: FAIR-EARTH: Land-Sea Binary Cross-Entropy Loss

| PE ↓ Resolution → | 5000 | 10000 | 15000 | 20000 |
|---|---|---|---|---|
| SPHERICAL WAVELET | $0.0177 \pm 0.0005$ | $0.0108 \pm 0.0002$ | $0.0095 \pm 0.0001$ | $0.0084 \pm 0.0001$ |
| SPHERICAL HARMONIC | $0.0158 \pm 0.0005$ | $0.0104 \pm 0.0005$ | $0.0089 \pm 0.0002$ | $0.0078 \pm 0.0003$ |
| THEORY | $0.0254 \pm 0.0028$ | $0.0144 \pm 0.0016$ | $0.0133 \pm 0.0008$ | $0.0114 \pm 0.0010$ |
| SPHEREM+ | $2.1551 \pm 0.0043$ | $2.2086 \pm 0.0012$ | $2.1889 \pm 0.0019$ | $2.2089 \pm 0.0015$ |
| SPHEREC+ | $2.1599 \pm 0.0032$ | $2.2055 \pm 0.0010$ | $2.1856 \pm 0.0018$ | $2.2074 \pm 0.0013$ |

Table 10: FAIR-EARTH: Carbon Emission Mean-Squared Error

| PE ↓ Resolution → | 5000 | 10000 | 15000 | 20000 |
|---|---|---|---|---|
| SPHERICAL WAVELET | $0.0306 \pm 0.0008$ | $0.0237 \pm 0.0001$ | $0.0169 \pm 0.0001$ | $0.0175 \pm 0.0002$ |
| SPHERICAL HARMONIC | $0.0269 \pm 0.0004$ | $0.0203 \pm 0.0004$ | $0.0158 \pm 0.0003$ | $0.0161 \pm 0.0003$ |
| THEORY | $0.0535 \pm 0.0060$ | $0.0270 \pm 0.0010$ | $0.0190 \pm 0.0006$ | $0.0194 \pm 0.0023$ |
| SPHEREM+ | $1.9559 \pm 0.0141$ | $1.8413 \pm 0.0019$ | $1.9528 \pm 0.0026$ | $1.8573 \pm 0.0007$ |
| SPHEREC+ | $1.9621 \pm 0.0086$ | $1.8433 \pm 0.0025$ | $1.9534 \pm 0.0024$ | $1.8578 \pm 0.0007$ |

Table 11: FAIR-EARTH: Surface Temperature Mean-Squared Error

| PE ↓ Resolution → | 5000 | 10000 | 15000 | 20000 |
|---|---|---|---|---|
| SPHERICAL WAVELET | $0.0993 \pm 0.0004$ | $0.0980 \pm 0.0007$ | $0.0986 \pm 0.0002$ | $0.0976 \pm 0.0006$ |
| SPHERICAL HARMONIC | $0.0984 \pm 0.0004$ | $0.0984 \pm 0.0006$ | $0.0982 \pm 0.0004$ | $0.0984 \pm 0.0005$ |
| THEORY | $0.1150 \pm 0.0067$ | $0.1162 \pm 0.0022$ | $0.1195 \pm 0.0014$ | $0.1201 \pm 0.0054$ |
| SPHEREM+ | $0.5981 \pm 0.0002$ | $0.5979 \pm 0.0001$ | $0.5981 \pm 0.0003$ | $0.5982 \pm 0.0001$ |
| SPHEREC+ | $0.5981 \pm 0.0001$ | $0.5986 \pm 0.0002$ | $0.5980 \pm 0.0002$ | $0.5983 \pm 0.0002$ |

Table 12: Land-Sea (Rußwurm et al. (2024)) Cross-Entropy Loss

| PE ↓ Resolution → | 5000 | 10000 | 15000 | 20000 |
|---|---|---|---|---|
| SPHERICAL WAVELET | $0.1622 \pm 0.0011$ | $0.1116 \pm 0.0007$ | $0.0848 \pm 0.0010$ | $0.0718 \pm 0.0002$ |
| SPHERICAL HARMONIC | $0.1631 \pm 0.0011$ | $0.1080 \pm 0.0010$ | $0.0773 \pm 0.0007$ | $0.0652 \pm 0.0011$ |
| THEORY | $0.2456 \pm 0.0124$ | $0.1796 \pm 0.0068$ | $0.1382 \pm 0.0055$ | $0.1265 \pm 0.0068$ |
| SPHEREM+ | $1.3853 \pm 0.0003$ | $1.3861 \pm 0.0001$ | $1.3862 \pm 0.0001$ | $1.3862 \pm 0.0000$ |
| SPHEREC+ | $1.3856 \pm 0.0002$ | $1.3861 \pm 0.0001$ | $1.3863 \pm 0.0000$ | $1.3862 \pm 0.0000$ |

Table 13: Checkerboard (Rußwurm et al. (2024)) Cross-Entropy Loss

Table 14: Surface temperature regression subgroup losses for various encodings. Coast (underlined) consistently exhibits greater losses.

| Subgroup | SW | SH | THEORY | SPHEREC+ |
|---|---|---|---|---|
| Land | 0.087 | 0.076 | 0.217 | 5.867 |
| Sea | 0.043 | 0.028 | 0.063 | 2.152 |
| Island | 0.049 | 0.041 | 0.047 | 2.101 |
| Coast | 0.101 | 0.083 | 0.249 | 5.835 |

| Encoding | Land-Sea Classification | | Surface Temperature Regression | |
|---|---|---|---|---|
| | **Best Country** (Value) | **Worst Country** (Value) | **Best Country** (Value) | **Worst Country** (Value) |
| SPHERICAL WAVELET | Finland (0.002) | Guyana (0.816) | Denmark (0.009) | Cambodia (0.232) |
| SPHERICAL HARMONIC | Honduras (0.001) | Spain (0.795) | Georgia (0.001) | Panama (0.333) |
| THEORY | Kyrgyzstan (0.001) | Spain (1.114) | Sierra Leone (0.002) | Vietnam (0.561) |
| SPHEREC+ | Austria (0.418) | Chile (1.257) | Romania (0.143) | Greenland (10.425) |

Table 15: Country discrepancies via FAIR-EARTH. Losses are respective to each dataset, and only countries with over 100 sampled points are included to mitigate noise. All encoding-dataset combinations exhibit a wide disparity in country-level performance.

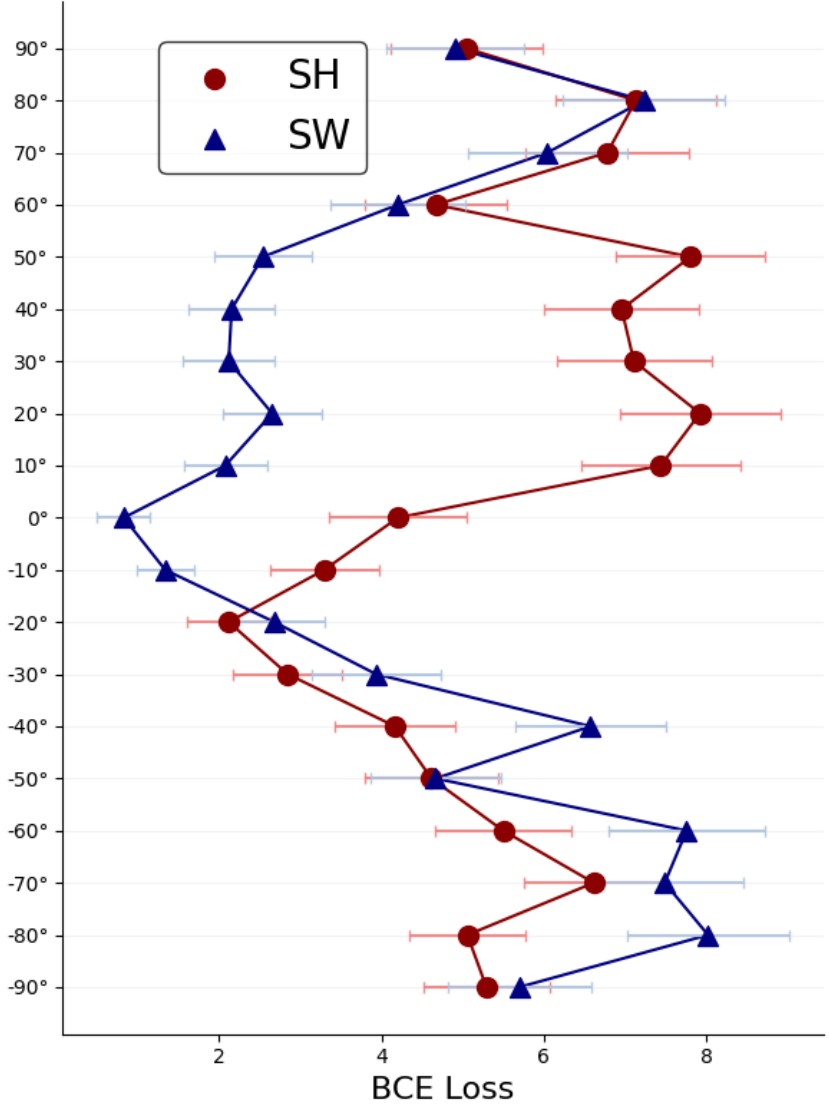

Figure 22: Differences in latitudinal loss on land-sea (Rußwurm et al. (2024)) classification, where means and error bars are based off of 1000 uniform samples at each latitude. SPHERICAL WAVELET performance clearly deteriorates as we move from the equator to the poles.

Table 16: Computational time for SPHERICAL HARMONIC and SPHERICAL WAVELET across different sizes, with faster times underlined. SPHERICAL WAVELET scales better than SPHERICAL HARMONIC, which can be attributed to its simpler formulation.

| | Generation Time (ms) | | Encoding Parameters | | |
|---|---|---|---|---|---|
| Size | SH | SW | Legendre Polyn. | Dilations | Rotations |
| 25 | $\underline{2.60 \pm 0.08}$ | $5.83 \pm 0.67$ | 5 | 1 | 25 |
| 100 | $\underline{16.44 \pm 0.72}$ | $22.67 \pm 2.85$ | 10 | 4 | 25 |
| 625 | $220.13 \pm 9.05$ | $\underline{129.58 \pm 18.43}$ | 25 | 5 | 125 |
| 900 | $368.73 \pm 17.93$ | $\underline{204.25 \pm 27.05}$ | 30 | 6 | 150 |

## A.5 WAVELET NOTES

**Computational Analysis** While wavelets are more heavily parameterized than their Fourier counterparts, one might expect them to be more computationally expensive to compute and tune. However, empirical evidence shows that hyperparameter searches converge at similar rates (Fig. 20), and SPHERICAL WAVELET demonstrates superior efficiency for large encodings (Table 16). A comparison of the computational requirements for SPHERICAL WAVELET and SPHERICAL HARMONIC encodings reveals a discrepancy in computational cost. Starting from our previous formulation, we define a family of valid wavelets as:

$$\{\psi_{a,\rho} \equiv \mathcal{R}(\rho)\mathcal{D}(a)\psi_M\}$$

where the dilation operator $\mathcal{D}(a)$ is defined as: $[\mathcal{D}(a)s](\omega) = [\lambda(a,\omega)]^{1/2}s(\omega_{1/a})$, with $\lambda$ being a scalar-valued operator.

The rotation operator $\mathcal{R}$ requires only a transformation into Euler space and three $3 \times 3$ matrix multiplications. Since the generation of rotation points is a one-time computation, this yields a computational complexity that scales linearly with encoding size.

In contrast, the closed-form SPHERICAL HARMONIC encoding with $L$ Legendre polynomials requires calculation of:

$$\sqrt{\frac{2l+1}{4\pi}\frac{(l-|m|)!}{(l+|m|)!}}P_l^m(\cos\lambda)$$

for all $m \leq l \leq L$. This higher computational burden makes SPHERICAL WAVELET comparatively efficient for larger encoding sizes. Additionally, computational and representational challenges of the factorials lead to instability at higher $L$ values (17), thus we restrict our analyses to $L = 30$.

**Non-Gabor Filters** We extended our investigation beyond Gabor wavelets to include preliminary experiments with spherical Butterfly and Mexican Hat wavelets, applying the same procedure with different mother wavelets. However, visualizing these wavelets' projections on the sphere suggests they are poorly suited for general Earth signals (Fig. 24). In our experiments on the FAIR-EARTH land-sea classification task, these alternatives performed worse than Gabor wavelets and even certain baseline encodings when controlling for all other parameters (Fig. 25). Future research directions could explore which signals these filters might be better suited for, and whether wavelet-related adjustments to INRs (Saragadam et al. (2023)) could address their current limitations.

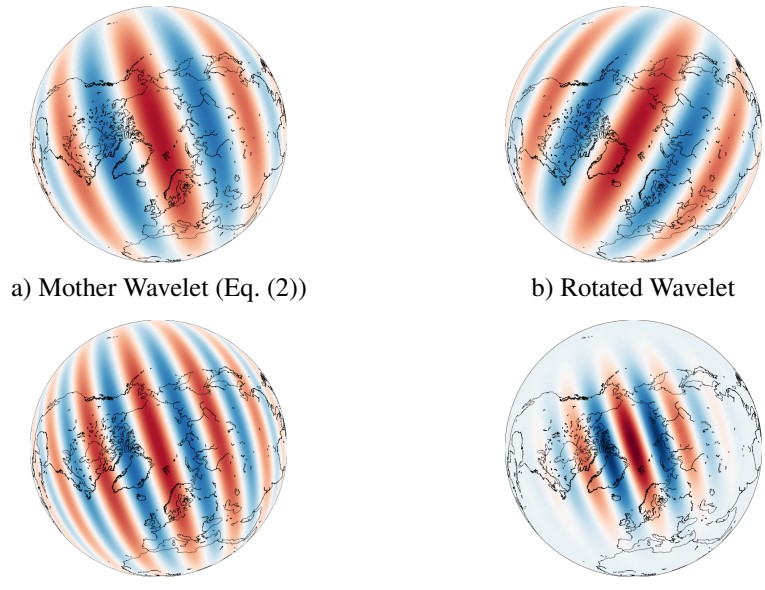

a) Mother Wavelet (Eq. (2))

b) Rotated Wavelet

c) Wave number controls central frequency of the Gaussian envelope

d) Dilation factor controls energy concentration

Figure 23: SPHERICAL WAVELET parameterization. All filters are identical to (Eq. (2)) except for one change in the specified parameter.

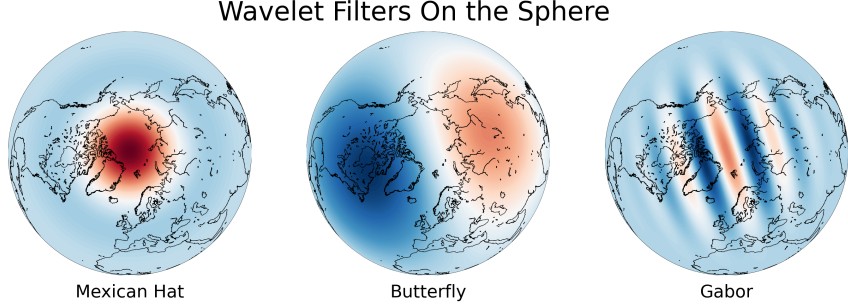

Wavelet Filters On the Sphere

Mexican Hat

Butterfly

Gabor

Figure 24: Visualizations of various wavelet filters on the sphere. Of the three, only Gabor consistently reaches state-of-the-art performance.

| Filter | Land-Sea | | Surface Temperature | |
|---|---|---|---|---|
| | 5000 | 10000 | 5000 | 10000 |
| Gabor (Ours) | $0.1419 \pm 0.0009$ | $0.0890 \pm 0.0003$ | $0.0306 \pm 0.0008$ | $0.0237 \pm 0.0001$ |
| Butterfly | $0.1943 \pm 0.0005$ | $0.1223 \pm 0.0003$ | $0.0523 \pm 0.0005$ | $0.0492 \pm 0.0010$ |
| Mexican Hat | $0.4001 \pm 0.0010$ | $0.3624 \pm 0.0021$ | $0.1550 \pm 0.0018$ | $0.1374 \pm 0.0009$ |

Figure 25: A comparison of different wavelet filter performance on the FAIR-EARTH land-sea classification task. Loss metric is respective to each dataset, and all other parameters are held constant.

