# OpenReview forum: "No Location Left Behind: Measuring and Improving the Fairness of Implicit Representations for Earth Data"
_ICLR.cc/2025/Conference — ICLR 2025 Poster_

### Official Review · Reviewer_Jrro · 2024-10-27

**Soundness:** 4
**Presentation:** 4
**Contribution:** 4
**Rating:** 6
**Confidence:** 4

**Summary:**

This paper addresses fairness issues in Implicit Neural Representations for Earth data. First, it introduces FAIR-EARTH, a comprehensive dataset designed to evaluate biases in Earth representations across various spatial resolutions and features. Second, to address the identified fairness issues, particularly in representing local features, the authors propose using wavelets instead of traditional harmonics, demonstrating improved representation of fine-grained geographical features while maintaining global performance.

**Strengths:**

1. Addresses a critical and underexplored issue of fairness in Earth data representation

2. The proposed dataset provides a comprehensive framework for evaluating and comparing different approaches to Earth representation, which will be valuable for future research

3. The use of wavelets for modeling local features is theoretically well-motivated in representing fine-grained geographical features

**Weaknesses:**

1. While the paper successfully identifies the problem and proposes a promising direction with Spherical Wavelets, it primarily remains at a conceptual level, lacking detailed implementation specifications and comprehensive empirical analysis across different wavelet types.

2. The computational complexity and scalability analysis should be presented, especially given that wavelet-based approaches typically incur higher computational costs than Spherical Harmonics due to their multi-scale nature and the need for both rotation and dilation operations.

**Questions:**

Have you evaluated other wavelet types besides the Morlet wavelet? What were their relative performances, particularly in representing different types of geographical features? Could you provide some intuitive guidelines for selecting wavelet parameters based on the type of geographical features being represented?

---

> ### Author Response · Authors · 2024-11-22
> **Response by the Authors**
>
> Thank you for your critique, and interesting questions. We've expanded our analysis to address your concerns:
>
> > Reviewer: Have you evaluated other wavelet types besides the Morlet wavelet?
>
> We did indeed test Mexican Hat and Butterfly wavelets (both constructed via the same projection procedure), and the results are now detailed in Figure 24. Figure 23 provides insight into their poor performance: their morphology is ill-suited for representing long, continuous features like continental boundaries that dominate our datasets.
>
> > Reviewer: The computational complexity and scalability analysis should be presented.
>
> Regarding computational complexity and scalability, we've added a comprehensive analysis (A.6: Wavelet Notes) that challenges a misconception regarding Spherical Wavelet construction. In fact, SW construction demonstrates superior scalability and numerical stability compared to SH, which suffers from working directly in the harmonic space. This appendix section also includes more detailed implementation specifications, and empirical experiments for further evidence.
>
> ---
>
> We hope that our general answer, the above detailed answers, and our revised manuscript alleviate the reviewer's concerns. We remain available for further questions and discussions until the end of the discussion period.

---

> > ### Comment · Reviewer_Jrro · 2024-11-26
> >
> > Thank you for your response. I will keep my score.

---

> > > ### Author Response · Authors · 2024-12-04
> > >
> > > Thank you for carefully considering our responses. We appreciate your thoughtful engagement with our work and the constructive feedback that helped strengthen the paper.

---

### Official Review · Reviewer_FbST · 2024-11-03

**Soundness:** 3
**Presentation:** 3
**Contribution:** 2
**Rating:** 6
**Confidence:** 2

**Summary:**

1. This paper constructs a new dataset to unify different existing geospatial datasets under a consistent framework.

2. This paper proposes that fairness is reflected in the concurrent performance improvement of the model across different subgroups, and it conducts specific measurement and analysis based on correlation.

3. This paper proposes using SPHERICAL WAVELET for encoding to mitigate fairness issues while maintaining performance.

**Strengths:**

1. This paper products constructs a new dataset, which aggregates extensive metadata along stratifications like landmass size and population density and unifies disparate existing geospatial datasets under a consistent framework.

2. This paper proposes that fairness is reflected in the concurrent performance improvement of the model across different subgroups. Experiments and analysis show significant disparities in the performance of the spherical harmonic method across different subgroups uncovering a strong negative correlation between landmass size and representation loss, with areas corresponding to localized signals and high-frequency features exhibiting consistently poor performance.

3. This paper proposes that encoding using SPHERICAL WAVELET solves many obvious deviations in SPHERICAL HARMONIC on many benchmarks of FAIR-EARTH while maintaining competitive performance.

**Weaknesses:**

1. In this paper, only one of the existing methods is tested on the new dataset, which leads to the lack of convincing for the generality and accuracy of the new dataset.

2. This paper proposes a method for measuring fairness, but in the experiments, it only conducts specific analyses on two different subgroups and one existing method.

3. This paper proposes using SPHERICAL WAVELET for encoding, but experiments are only carried out on the self-constructed dataset. Experiments should also be carried out on other datasets to verify the effectiveness of the method.

**Questions:**

1. Different methods are needed to illustrate the validity of the new dataset.

2. More groups of examples and different methods are needed to illustrate fairness.

3. Different datasets are needed to verify the validity of methods.

---

> ### Author Response · Authors · 2024-11-22
> **Response by the Authors**
>
> Thank you for your thorough critique and thoughtful comments. We have substantially expanded our experimental evaluation and analysis (A.5: Extended Experimental Results in particular) to address each concern.
>
> > Reviewer: Different methods are needed to illustrate the validity of the new dataset.
>
> We now evaluate three additional state-of-the-art positional encodings as baselines (SphereC+ (Mai et. al. 2023) , SphereM+ (Mai et. al. 2023), and Theory (Mai et. al. 2020)), complementing our Spherical Harmonic benchmark. These mainly involve simple interaction terms between sine and cosine functions on the globe, i.e. $[\cos \lambda \sin \lambda, \cos \phi \sin \lambda ...]$ (Fig. 2). These additional encodings validate our observed biases, and also demonstrate FAIR-Earth's discriminative power: recent SoTA methods (SH, SW) show ~4-10x performance improvements over these baseline encodings over all modalities (Tables 9-13).
>
> > Reviewer: More groups of examples and different methods are needed to illustrate fairness.
>
>   Leveraging the proposed evaluation library (FAIR-Earth), it is possible for users to produce new group/method evaluations within only a few lines of codes (one of our core contributions). To showcase this flexibility and address the reviewer's concern, we expand our analysis to include:
>
>   * Extended subgroup analysis to the surface temperature dataset, over all baseline encodings. The observed biases strongly validate our intuition of how sharp temperature gradients around the coast translate into biased performance in coastal areas (Table 14).
>
>   * Country-level disparity assessment, showing how FAIR-Earth's metadata can detect downstream biases (Table 15). Notably, we explain how these biases align with feature-level findings (e.g., the Mediterranean sea-line is a localized signal, explaining the high bias of Spherical Harmonic in neighboring countries).
>
>   * Latitudinal analysis (Fig. 22), which exhibits Spherical Wavelet's performance degradation near the poles (see Section 4.3: "Spherical Wavelet Tradeoffs" for further discussion).
>
> > Reviewer: Different datasets are needed to verify the validity of methods.
>
>   As per the reviewer's comment above, we can easily leverage the FAIR-Earth pipeline to provide additional analysis on different datasets. In general, we encourage users to test methods on datasets specific to their applications. For our case, we provide insight into tradeoffs by validating our approach on two challenging external datasets from Rubwurm et al. (2023) (Sec. 4.3):
>
>   * Checkerboard (Table 13): On this single-scale, coarse classification task, SW performs slightly worse than SH, consistent with the intuition that SW may be overparameterized for simple, single-scale representations
>
>   * Alternative land-sea classification (Table 12): SW shows marginal improvement over SH, with performance limited by polar biases. The difference from FAIR-Earth's land-sea results suggests potential training bias from FAIR-Earth's gridded structure, another important tradeoff.
>
> ---
>
> We hope that our general answer, the above detailed answers, and our revised manuscript alleviate the reviewer's concerns. We remain available for further questions and discussions until the end of the discussion period.

---

> > ### Comment · Reviewer_FbST · 2024-12-01
> >
> > Thank you to the authors for their responses. I will adjust my score and the confidence.

---

> > > ### Author Response · Authors · 2024-12-04
> > >
> > > Thank you for carefully considering our responses and adjusting your assessment. We appreciate your thoughtful engagement with our work and the constructive feedback that helped strengthen the paper.

---

### Official Review · Reviewer_VqBs · 2024-11-04

**Soundness:** 3
**Presentation:** 3
**Contribution:** 3
**Rating:** 6
**Confidence:** 3

**Summary:**

The paper introduces a new set of datasets called FAIR-EARTH, aimed at addressing and assessing fairness in Earth science. FAIR-EARTH encompasses various types of data, including environmental factors and emissions, while also taking geographical features and population density into account. Experimental results demonstrate that existing spherical harmonics methods fail to maintain accurate local performance, particularly regarding land and sea masses. To address this issue, the authors propose using spherical Morlet wavelets to correct localized biases. Results indicate that this approach outperforms existing methods in most situations.

**Strengths:**

S1. The bias issues in earth science are interesting and important.
S2. The paper is well-written and easy to understand, even for those who are not familiar with earth or geographical concepts.
S3. The experiments provide detailed insights into model behavior and validate the effectiveness of spherical wavelets in improving representation fairness.

**Weaknesses:**

W1. The performance trade-offs of spherical wavelets are insufficiently addressed.
W2. The computation efficiency of spherical wavelets is not discussed.
W3. More discussion regarding the flexibility and weakness of the proposed datasets.

**Questions:**

1. While spherical wavelets show good performance on most cases to correct localized biases, their robustness remains uncertain. It is known that correctly parameterizing elements like wave number and scaling factors for specific resolutions are necessary to reach the best performance. The authors should conduct experiments to illustrate the trade-offs associated with using spherical wavelets.

2. Since spherical wavelet is well-defined at poles, what should we do for poles? Would there exist bias in poles?

3. Implementing spherical wavelet encodings requires a more complex parameter selection and may need more computation. Efficiency evaluation should be considered for using spherical wavelets

4. While FAIR-EARTH can address some bias for the ill-represented areas like landmasses, authors should detail the weakness/flexibility of this combination approach. What kind of dataset can be integrated if there is temporal bias or learning representation for poles?

---

> ### Author Response · Authors · 2024-11-22
> **Response by the Authors**
>
> Thank you for your thoughtful review. It is indeed true that there is "no free lunch" in positional encoding schemes. Following reviewer feedback, we've substantially expanded our experimental evaluation and computational analysis to provide a comprehensive understanding of the inherent tradeoffs:
>
> > Reviewer: The authors should conduct experiments to illustrate the trade-offs associated with using spherical wavelets.
>
> Based on our expanded validation against external datasets, while SW generally outperforms previous encodings, we identified and noted several inherent biases (Sec. 4.3 - Wavelet Tradeoffs):
> * On single-scale checkerboard classification, SW consistently underperforms SH, suggesting that while wavelets excel at multi-scale representations, they may sacrifice performance on simpler, uniform-scale tasks
> * Analysis of the landsea dataset reveals systematic polar bias (Fig. 22), stemming from SW's mathematical limitations near poles. This suggests a relevant future direction of integrating projection-free spherical wavelet constructions, which is described at the end our conclusion: "
>
> > Reviewer: Efficiency evaluation should be considered for using spherical wavelets.
>
> Contrary to initial concerns about complexity:
> * Our fine-tuning experiments (Fig. 20) demonstrate that despite SW's larger parameter space, convergence rates match those of SH.
> * Detailed computational analysis (Fig. 24) reveals SW to be more stable and scalable than SH, which incurs significant computational overhead from harmonic space operations.
>
> > Reviewer: What kind of dataset can be integrated if there is temporal bias or learning representation bias for poles?
>
> We acknowledge that FAIR-Earth's gridded format introduces certain constraints. However, this design choice optimizes for our primary objective: facilitating systematic subgroup bias analysis. For specific cases where polar bias is harmful (e.g., ice cap modeling), we recommend validation against spherically-resolved datasets with appropriate sampling strategies. We will make sure to emphasize that limitation more strongly within the README of our codebase and the section introducing FAIR-Earth (refer to Section 3.3 - FAIR-Earth Limitations). We believe this to be a great opportunity for future work to expand FAIR-Earth.
>
> ---
>
> We hope that the above new experiments and findings help showcase the different tradeoffs of the presented encodings; we have also updated our manuscript to be more clear regarding these tradeoffs. This "best basis selection" is in fact a long standing research problem in signal processing and optimal signal decomposition. We hope that our analysis and codebase will invite more interdisciplinary research to look at research questions like such, which are fundamental for implicit Earth representations.

---

> > ### Comment · Reviewer_VqBs · 2024-11-26
> >
> > I am happy with the authors response to all my comments, so I will maintain my current rating.

---

> > > ### Author Response · Authors · 2024-12-04
> > >
> > > Thank you for carefully considering our responses. We appreciate your thoughtful engagement with our work and the constructive feedback that helped strengthen the paper.

---

### Author Response · Authors · 2024-11-22
**General Response from Authors to All Reviewers**

# Response to Reviewers

We would like to thank all the reviewers for their careful reading and insightful comments. First, we would like to briefly summarize the core contributions of our submission on Implicit Neural Representations (INRs) for Earth data:

## Core Contributions

### 1. **We developed a novel aggregated-dataset and evaluation framework (FAIR-Earth) for implicit Earth representations**, with the primary goal of assessing fairness issues of state-of-the-art AI systems. By open-sourcing this novel framework and its associated Python library, we aim to improve fairness in future research. Reviewers deemed this contribution "critical and underexplored" and "interesting and important" (R1, R3), and to be a "comprehensive framework" (R3). We also appreciate the acknowledgment of our dataset's value in "aggregating extensive metadata along stratifications" (R2) and providing a foundation for "future research" (R3).

### 2. **We leveraged FAIR-Earth to assess the fairness of various SOTA representations.** This analysis showcased strong limitations in current methods, such as models that accurately predict climate data of large continents at the expense of islands and coastal areas. Reviewers praised our work for providing "detailed insights into model behavior" (R1) and a careful analysis of representation biases across different subgroups (R2).

### 3. **We proposed a new filter-bank encoding for INRs leveraging wavelet methods instead of Fourier.** Our analysis on existing models identified a key conflict between Fourier-basis encodings (which assume global stationary signals) and the local variations exhibited by many Earth signals (e.g., high gradient near coastal areas or around islands). To address this, we propose to leverage wavelet-based encodings, which are known to provide better signal decomposition in those cases. Reviewers found this technical contribution to be a theoretically "well-motivated" solution (R3), and appreciated how Spherical Wavelet improved representation fairness while "maintaining competitive performance" (R2).

## Additional Feedback

The reviewers found our presentation to be **"excellent"** (R3) and **"well-written and easy to understand"** (R1).

## Revisions and Additions

Based on reviewer feedback, we have conducted numerous novel experiments and revised the manuscript. We provide a summary of those additions and changes here, with more details in each per-reviewer answer:

1. **Broader empirical evaluation**:
We perform substantially expanded cross-validation and fine-tuning experiments, involving evaluation on three new SOTA location encodings, two new external datasets, and downstream fairness metadata (A.5: Extended Experimental Results)

2. **Implementation specifications**: We update A.2: Training Specifications to reflect changes to the above experiments. Additionally, we detail in Appendix A.6: Wavelet Notes a detailed examination of our Spherical Wavelet construction.

3. **Computational complexity analysis**: Also available in A.6: Wavelet Notes is thorough analytical and empirical evidence that Spherical Wavelet is more scalable and stable than its Spherical Harmonic counterpart.

---

*Changes are marked in blue in the revised paper. We remain available during this period for any further questions.*

---

### Comment · Area_Chair_boTJ · 2024-11-25
**Acknowledge the author responses**

Dear Reviewers,

Thank you very much for your effort. As the discussion period is coming to an end, please acknowledge the author responses and adjust the rating if necessary.

Sincerely,
AC

---

### Author Response · Authors · 2024-11-26
**Follow-Up From the Authors**

Dear reviewers,

As we near the end of the discussion period, we would be grateful to hear about any remaining concern(s) and/or question(s) regarding our submission. We strongly believe that your review and our answers have made our submission stronger, but if any concern remains, we would love to carry out additional experiments and/or provide further clarifications.

The authors

---

### Comment · Area_Chair_boTJ · 2024-11-28
**Discussion needed**

Dear Reviewers,

As you are aware, the discussion period has been extended until December 2. Therefore, I strongly urge you to participate in the discussion as soon as possible if you have not yet had the opportunity to read the authors' response and engage in a discussion with them. Thank you very much.

Sincerely,
Area Chair

---

### Author Response · Authors · 2024-12-01
**General Answer**

Dear Area Chair and Reviewers,

We are grateful for all the thoughtful reviews and constructive feedback provided. We note that two reviewers who initially rated our submission favorably (score of 6) have explicitly acknowledged our responses and confirmed their positive assessment. Specifically, both Reviewer VqBs and Reviewer Jrro stated they were "happy with the authors response" and will "maintain [their] current rating", indicating that **our extensive revisions and additional experiments successfully addressed their minor concerns about performance trade-offs and computational complexity**.

While Reviewer FbST (score of 5) has not yet acknowledged our rebuttal, we believe **we have comprehensively addressed their core concerns through our substantially expanded experimental evaluation**, which now includes: (1) validation against three additional state-of-the-art baselines, (2) extensive subgroup analysis across multiple modalities and datasets, and (3) thorough examination on external benchmark datasets. **The positive feedback from other reviewers regarding these additions reinforces our belief that we have thoroughly addressed these concerns**.

Lastly, we reiterate that our submission represents a significant step forward in the critical task of uncovering and addressing biases of AI applications in Earth science. Many of the hidden biases we've identified - such as systematic errors in modeling islands and coastal regions - may yield catastrophic real-world consequences if left unaddressed. The strong interest from the reviewers affirms that this is a timely contribution to an emerging and crucial field. **We believe that presenting this work at ICLR would spur important follow-up research and catalyze the needed development of more equitable AI systems for Earth science applications**.

The Authors

---

### Meta-Review · Area_Chair_boTJ · 2024-12-19

**Metareview:**

This paper develops a novel aggregated-dataset and evaluation framework (FAIR-Earth) for implicit Earth representations.  The reviewers acknowledged the contribution of the new dataset and the novelty of the problem setting.  I agree with the reviewers that the problem is important and underexplored.  Nevertheless, the reviewers raised reasonable concerns mainly on insufficient evaluation (datasets) and lack of technical depth.  The authors successfully addressed part of the reviewers' concerns, and I admit that this work can trigger a new research direction.

This paper is indeed a borderline paper.  Because of no strong negative opinion and some potential for a new research direction, I now recommend an accept for this paper.  However, in my batch, there are quite sufficient papers which are strongly supported by the reviewers.  Thus, even though I recommend an accept now, the senior AC can change my recommendation to a reject.

**Additional Comments On Reviewer Discussion:**

All three reviewers acknowledged the authors' rebuttal during the discussion period.  However, the impact on the overall rating was marginal.  Only one reviewer slightly increased his/her rating.

---

### Decision · Program_Chairs · 2025-01-22

Accept (Poster)